# Cognitive decline in diabetic mice predisposed to Alzheimer's disease is greater than in wild type

Marta Carús-Cadavieco[1,*], Inés Berenguer López[1,*], Alba Montoro Canelo[1,2], Miguel A Serrano-Lope[1], Sandra González-de la Fuente[3] , Begoña Aguado[3] , Alba Fernández-Rodrigo[1] , Takaomi C Saido[4], Ana Frank García[5], César Venero[6] , José A Esteban[1], Francesc Guix[1,7], Carlos G Dotti[1] 

**In this work, we tested the hypothesis that the development of dementia in individuals with type 2 diabetes (T2DM) requires a genetic background of predisposition to neurodegenerative disease. As a proof of concept, we induced T2DM in middle-aged hAPP NL/F mice, a preclinical model of Alzheimer's disease. We show that T2DM produces more severe behavioral, electrophysiological, and structural alterations in these mice compared with wild-type mice. Mechanistically, the deficits are not paralleled by higher levels of toxic forms of Aβ or by neuroinflammation but by a reduction in γ-secretase activity, lower levels of synaptic proteins, and by increased phosphorylation of tau. RNA-seq analysis of the cerebral cortex of hAPP NL/F and wild-type mice suggests that the former could be more susceptible to T2DM because of defects in trans-membrane transport. The results of this work, on the one hand, confirm the importance of the genetic background in the severity of the cognitive disorders in individuals with T2DM and, on the other hand, suggest, among the involved mechanisms, the inhibition of γ-secretase activity.**

## Introduction

Diabetes is a complex metabolic disorder characterized by increased levels in blood glucose because of an altered insulin production by pancreatic cells (type 1 diabetes) or by impaired insulin response (type 2, abbreviated T2DM), although in later stages, is also accompanied by altered insulin production. In addition to blood hyperglycemia, T2DM patients also present hyperinsulinemia and peripheral insulin resistance (Wu & Parhofer, 2014). T2DM is generally accompanied by comorbidities such as obesity, peripheral inflammation, dyslipidemia or hypertension,

constituting a syndrome known as metabolic syndrome (MS) (Huang, 2009). These morbid states are commonly associated with vascular complications affecting kidney function (nephropathy), vision (retinopathy), and peripheral sensations (peripheral neuropathy) (Donaghue et al, 2009). In addition, a considerable percentage of diabetic patients also show structural and functional changes in the central nervous system, including, in general subtle, defects in information processing, psychomotor efficiency, attention, and cognitive flexibility (Brands et al, 2005).

Although initial population-based studies showed some contradictory results, the effect of T2DM on pathologic cognitive decay, including higher risk of developing Alzheimer's disease (AD), has been established. Thus, The Rotterdam study pioneered the concept that T2DM represents a twofold increase in risk of dementia, mainly Alzheimer's and vascular dementia (Ott et al, 1999). Since then, many other studies worldwide have consistently shown an increased risk to develop dementia in the T2DM population. For example, the Rochester study showed a 1.6fold increased risk of all types of dementia (Leibson et al, 1997). Other large population-based studies confirmed that diabetes increased the risk of AD (Peila et al, 2002; Huang et al, 2014). In any case, not all individuals with diabetes develop late-onset AD (LOAD), confirming the need of summative factors for the occurrence of cognitive complications.

Numerous studies based on the use of animal models of diabetes have demonstrated (although there are highly variable results) that diabetes induces alterations in mitochondrial function, in oxidative stress, in inflammation, cellular proteostasis, and cellular metabolism (Studzinski et al, 2009; Morrison et al, 2010; Arnold et al, 2014; Petrov et al, 2015) that could easily explain how this metabolic alteration leads to cognitive deficits. Other studies have analyzed the effect of diabetes in mice with a strong background of hereditary AD (e.g., the triple transgenic mouse), concluding that diabetes produce significant changes in the brain to produce AD (Pedersen & Flynn, 2004; Knight et al, 2014; Vandal et al, 2015;

[1]Molecular Neuropathology Unit, Physiological and Pathological Processes Program, Centro de Biología Molecular Severo Ochoa(CBM), CSIC-UAM, Madrid, Spain [2]Escuela Técnica Superior (E.T.S) de Ingeniería Agronómica, Alimentaria y de Biosistemas, Universidad Politécnica de Madrid, Madrid, Spain [3]Genomics and NGS Facility, Centro de Biología Molecular Severo Ochoa(CBM) CSIC-UAM, Madrid, Spain [4]Laboratory for Proteolytic Neuroscience, RIKEN Center for Brain Science, Saitama, Japan [5]Department of Neurology, Division Neurodegenerative Disease, University Hospital La Paz, Madrid, Spain [6]Department of Psychobiology, Universidad Nacional de Educación a Distancia, Madrid, Spain [7]Department of Bioengineering, Institut Químic de Sarrià (IQS) – Universitat Ramón Llull (URL), Barcelona, Spain

Correspondence: francesc.guix@iqs.url.edu; cdotti@cbm.csic.es
*Marta Carús-Cadavieco and Inés Berenguer López are first authors

Velazquez et al, 2017). However, the models used in these studies (for example, the triple transgenic is based on the over-expression of human amyloid precursor protein with Swedish mutation, presenilin-1 mutation [PS-1M146V] and tau P301L) taught us that diabetes worsens the Alzheimer's phenotype but not whether or not an Alzheimer's predisposed genotype is required. Therefore, we have now carried out a behavioral, electrophysiological, biochemical, histological, and transcriptomic study of the effect of diabetes on 14–15 mo old WT and Alzheimer´s genetically predisposed hAPP NL/F mice (Saito et al, 2014).

hAPP NL/F mice are knockin mice of the humanized amyloid beta-peptide (Aβ) region with Swedish (NL) and Iberian (F) mutations, resulting in high levels of Aβ42 production starting at 2–3 mo of age, with plaques evident after 12 mo and mild behavioral phenotype at later ages (Sasaguri et al, 2017). We therefore consider these mice a suitable model for studying the weight of genetic and environmental influences on the development of Alzheimer's disease.

Our results show that these mice develop signs and symptoms of cognitive pathology when suffering from chronic T2DM that are more severe than non-AD predisposed animals. Furthermore, other results of this work indicate that part of the greater deficits in this genetic background is because of T2DM-mediated inhibition of γ-secretase activity.

# Results

## Combined chronic high-fat diet (HFD) and low doses of streptozotocin (STZ) lead to signs of peripheral T2DM in both WT and hAPP NL/F adult mice

We induced T2DM in WT and hAPP NL/F mice using a combination of HFD and STZ.

Although STZ is a compound commonly used as an experimental model of type 1 diabetes (because this antibiotic induces the destruction of pancreatic β cells leading to hypoinsulinemia and hyperglycemia), its use at low doses and in combination with HFD allows it to mimic the final stages of T2DM (Furman, 2015). Mice between 7 and 9 mo of age were subjected to HFD for 14 wk, at which time, an i.p. injection of STZ (40 mg/kg) per day was performed for five consecutive days. After this treatment, HFD was continued for another 6 wk (age of the mice at the end of the experiment was 12–14 mo). Weight and blood glucose levels were controlled every 2 wk; glucose and insulin resistance tests were performed during weeks 16–18. Behavioral tests commenced on week 18 and lasted for 2 wk. Electrophysiological tests were carried out few days after the behavioral tests, always with the mice fed with control or HFD. A scheme of the experimental paradigm is shown in Fig 1. Experimental details can be found in the Materials and Methods section.

Mice subjected to the HFD and STZ with WT or hAPP NL/F backgrounds developed classical signs of metabolic syndrome compared with mice with same background under control treatment, including weight gain (Fig 2A) and hyperglycemia (Fig 2B). Glucose tolerance test (GTT) revealed a significant impairment in blood glucose clearance after the i.p. injection of a highly concentrated glucose solution (Fig 2C). Relatedly, the insulin tolerance test (ITT) revealed a significant difference in glucose levels in response to an i.p. insulin injection

compared with mice under control diet (Fig 2D), a behavior assumed to reveal insulin resistance (Bowe et al, 2014). To test if the paradigm also resulted in brain insulin resistance, we next analyzed the levels of phosphorylation of the insulin receptor substrate 1 (IRS1), an adapter protein of the insulin receptor with the capacity to either stimulate insulin signaling or to inhibit it depending on the phosphorylation state (Yaribeygi et al, 2019; James et al, 2021). Consistent with brain insulin resistance, the T2DM hAPP NL/F mice but not WT mice showed a significant increase in IRS1 phosphorylated in Serine 307 (Fig S1A and B). This last result implies that the brain of these AD genetically predisposed mice is more susceptible to the metabolic disturbances that occur in the course of type 2 diabetes, probably as a consequence of defects in plasma membrane permeability (see below).

## T2DM impairs spatial learning and memory, more in hAPP NL/F mice than in WT

A number of clinical studies have revealed mild mental performance deficits in particular patients with long-term T2DM compared with age-, sex-, and education-matched controls, especially on measures of verbal memory, information processing speed and attention, and executive functioning (Wahlin et al, 2002; Brands et al, 2007). Because the latter can be evaluated in mice with relatively simple behavioral tests, we next exposed our control and T2DM mice to the Open Field test, the Novel Object Recognition test, the Barnes maze test, and the Y-maze test.

The Open Field test (OF) is often used for assessing anxiety and willingness to explore. The experimental design is schematically represented in Fig S2A. We observed that T2DM condition induced a decrease of the mobility only in the hAPP NL/F background (Fig 3A–C). We should note that both WT and hAPP NL/F mice had the same level of motor activity under control diet. Therefore, the decreased mobility in the hAPP NL/F diabetic mice may be because of a higher anxiety/reduced motivation.

The Novel Object Recognition (NOR) task evaluates the use of learning and recognition memory. It is widely accepted that in both the monkey and the rat brains, the cortex (perirhinal and para-hippocampal regions) and hippocampus play an important role in object recognition memory. In neither genetic background did T2DM affect the time the mice spent exploring the new object compared with the total time exploring both objects (Fig S2B). The total exploration time (both objects) was also not significantly affected by T2DM in either genetic background (Fig S2C). These results indicate that T2DM does not significantly modify the neural circuits responsible for this type of behavior, not even in individuals genetically predisposed to AD.

The Barnes maze is a behavioral test developed, and frequently used, for assessing spatial learning and memory (Barnes, 1979; Pitts, 2018). It represents a well-established alternative to the more popular Morris Water maze and offers the advantage of being free from the potentially confounding influence of swimming behavior, especially relevant for mice. Like the Morris water maze, the Barnes maze is a hippocampal-dependent task where animals learn the relationship between distal cues in the surrounding environment and a fixed escape location. We carried out four trials per day and measured the time it took each mouse to find the escape hole each day. A schematic representation of the experimental design can be

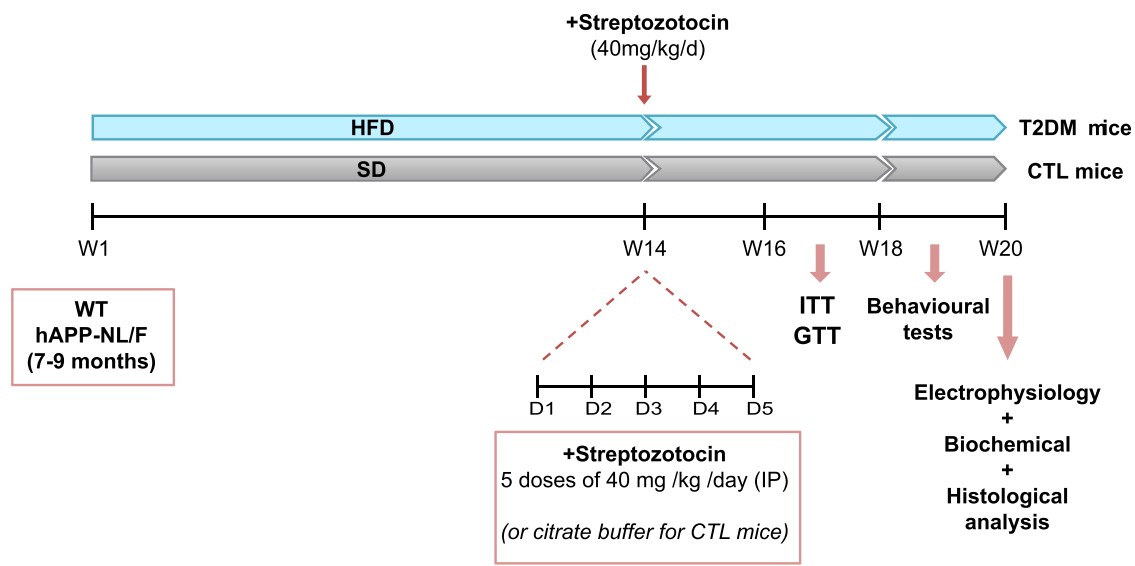

**Figure 1.  Schematic representation of the experimental design.**

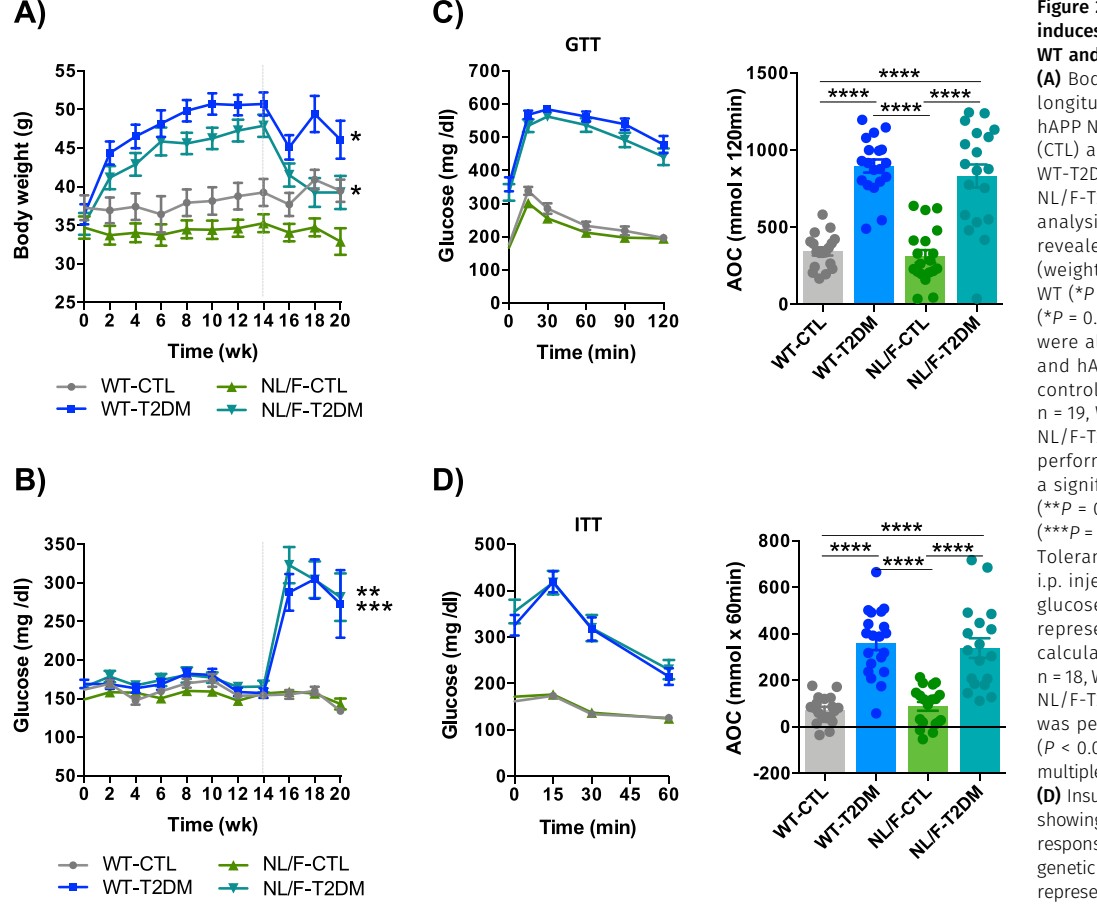

**Figure 2.  Combination of HFD+STZ induces peripheral T2DM signs in both WT and hAPP NL/F mice.**
**(A)** Body weight was measured longitudinally every 2 wk in WT and hAPP NL/F knockin (NL/F) mice, control (CTL) and T2DM (WT-CTL n = 19, WT-T2DM n = 20, NL/F-CTL n = 20, NL/F-T2DM n = 20). **(B)** Statistical analysis performed by unpaired *t* test revealed a significant effect of T2DM (weight difference week 20–week 0) in WT (*$P$ = 0.0124), and in NL/F mice (*$P$ = 0.0102), (B) Blood glucose levels were also monitored every 2 wk in WT and hAPP NL/F knockin (NL/F) mice, control (CTL), and T2DM, (WT-CTL n = 19, WT-T2DM n = 20, NL/F-CTL n = 20, NL/F-T2DM n = 20). Statistical analysis performed by unpaired *t* test revealed a significant effect of T2DM in WT (**$P$ = 0.0017), and in NL/F mice (***$P$ = 0.0005), at week 20. **(C)** Glucose Tolerance Test (left graph) after an i.p. injection of a highly concentrated glucose solution. The right bar plot represents the area of the curve calculated from the left graph (WT-CTL n = 18, WT-T2DM n = 19, NL/F-CTL n = 19, NL/F-T2DM n = 19). Statistical analysis was performed by one-way ANOVA ($P$ < 0.0001), followed by Tukey's multiple comparisons test (****$P$ < 0.0001). **(D)** Insulin Tolerance Test (left graph) showing differences in glucose levels in response to an insulin injection in both genetic backgrounds. The right bar plot represents the area of the curve calculated from the left graph (WT-CTL n = 19, WT-T2DM n = 20, NL/F-CTL n = 19, NL/F-T2DM n = 18). Statistical analysis was performed by one-way ANOVA ($P$ < 0.0001), followed by Tukey's multiple comparisons test (****$P$ < 0.0001). All plots show mean values ± SEM. Dotted lines indicate STZ injection timepoint.

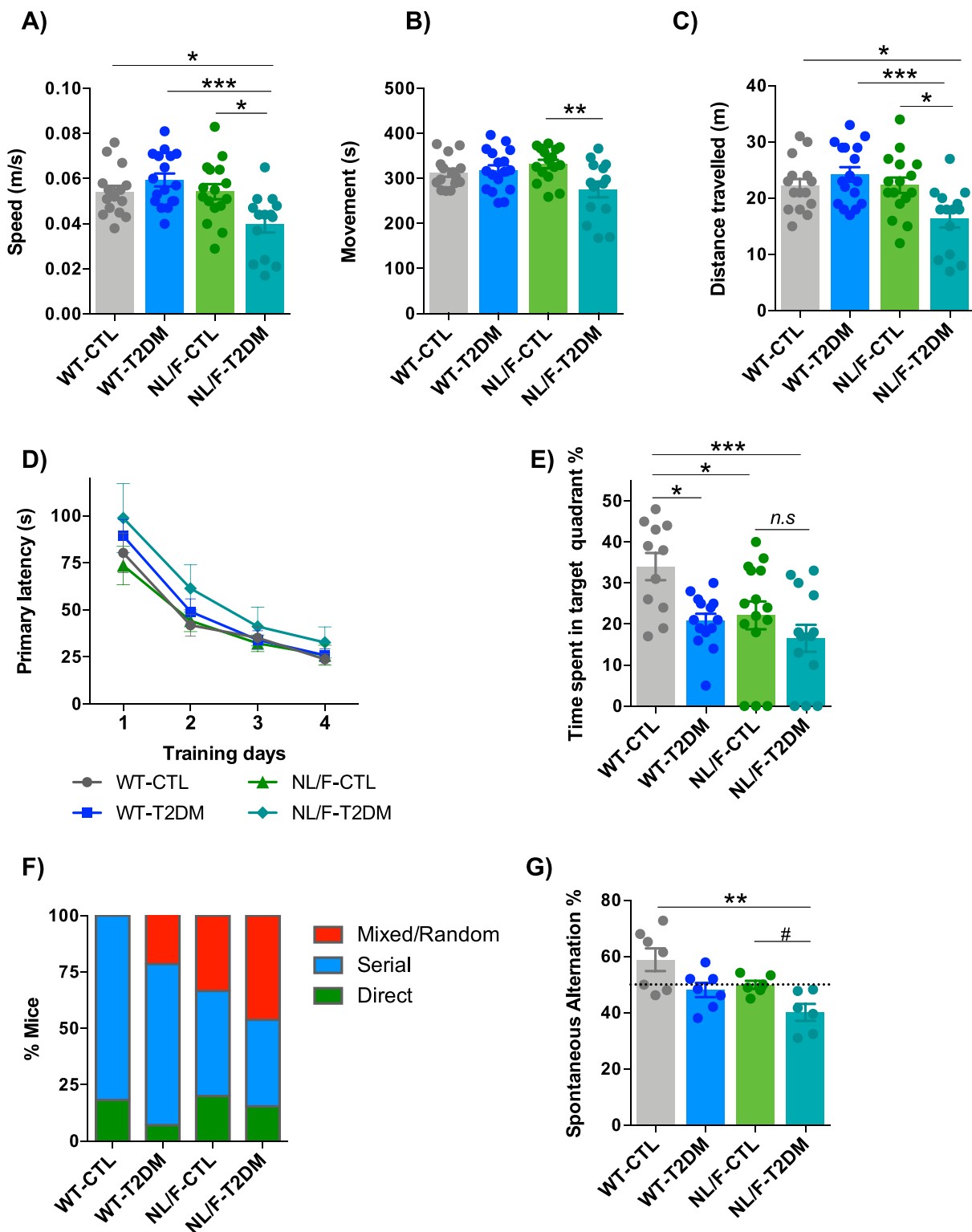

**Figure 3. T2DM affects behavior, more in hAPP NL/F mice than in WT.**
**(A, B, C)** Open Field paradigm. **(A)** The graph shows the average speed during the Open Field paradigm of WT and hAPP NL/F knockin (NL/F) mice, control (CTL), and T2DM (WT-CTL n = 15, WT-T2DM n = 17, NL/F-CTL n = 16, NL/F-T2DM n = 14). Statistical analysis was performed by one-way ANOVA ($P$ = 0.0006), followed by Tukey's multiple comparisons test (*$P$ < 0.05, ***$P$ < 0.001). **(B)** The graph shows the time the mice spent moving during the Open Field paradigm, WT and hAPP NL/F knockin (NL/F) mice, control (CTL), and T2DM (WT-CTL n = 15, WT-T2DM n = 17, NL/F-CTL n = 16, NL/F-T2DM n = 14). Statistical analysis was performed by one-way ANOVA ($P$ = 0.0132), followed by Tukey's multiple comparisons test (**$P$ < 0.01). **(C)** The graph shows the distance travelled by the mice during the Open Field paradigm, WT and hAPP NL/F knockin (NL/F) mice, control (CTL), and T2DM (WT-CTL n = 15, WT-T2DM n = 17, NL/F-CTL n = 16, NL/F-T2DM n = 14). Statistical analysis was performed by one-way ANOVA ($P$ = 0.0011), followed

found in Fig S2D. Fig 3D shows that the time spent to find the escape hole was not altered by T2DM in WT mice. In contrast, this learning process was, although not significant, slower in the hAPP NL/F mice with T2DM (Fig 3D). The last day of the test, we measured the time mice spent on the target quadrant where the escape hole is located. T2DM per se decreases the time spent on the target quadrant in WT mice (Fig 3E), in agreement with previous studies (Anderson et al, 2014). This time was significantly shorter in hAPP NL/F mice as compared with WT mice under control diet, implying a critical effect of the Alzheimer's genetic background. Furthermore, there was a trend for T2DM to decrease it even further, although this effect was not statistically significant (Fig 3E). T2DM also affected the strategy to find the escape hole at the end of the training period in a genetic background-dependent manner. Typically, depending on how well they have learned, at the end of the training period, mice can reach the escape hole through direct searching, sequential exploration of the holes or random exploration, reflecting, respectively, better to worse learning. Most WT mice used either the direct (2 out of 11 under control diet and 1 out of 14 with T2DM) or sequential (9 out of 11 under control diet and 10 out of 14 with T2DM) path to the hole. On the contrary, a significant number of hAPP NL/F mice with T2DM used the random search strategy: 6 out of 13 compared with 5 out of 15 in the control diet (Fig 3F). Furthermore, T2DM tended to increase the distance the mice travelled to reach the escape hole, both in the WT and the hAPP NL/F mice (Fig S2E). Differently, the number of errors the mice made before locating the escape hole was increased by T2DM in the hAPP NL/F mice (Fig S2F). Therefore, the genetic background of predisposition to AD slightly perturbs spatial learning and memory, and this disturbance is exacerbated by T2DM.

The Y-maze is a test that investigates spatial learning and short-term memory in mice (Kraeuter et al, 2019). This process is thought to require interaction across several different regions of the brain, such as the hippocampus and prefrontal cortex. Although T2DM affected alternation, the effect was significant only in the hAPP NL/F mice (Fig 3G). It is worth noting that control mice with hAPP NL/F background have a lower level of alternation than WT and that the T2DM lowers it further, significantly.

These three behavioral tests allow us to conclude that, T2DM affects certain patterns of cognitive and mood processes, which are more pronounced in individuals with an innate predisposition to AD, the most notable being reduced working memory (i.e., the type of memory that is needed for the execution of cognitive tasks).

### T2DM impairs basal synaptic transmission in hAPP NL/F mice, not in WT

The behavioral changes induced by diabetes in hAPP NL/F mice may reflect changes in synaptic activity of hippocampal circuits.

Therefore, we next analyzed the effect of T2DM on CA3-to-CA1 synaptic transmission and synaptic plasticity (long-term potentiation). Acute hippocampal slices were prepared 20 wk after the beginning of the HFD and 6 wk after the STZ injection (see Fig 1).

T2DM did not alter basal synaptic transmission, as measured in field recordings (fEPSPs) in response to increasing stimulus intensities, in WT mice (Fig 4A, left panel). In contrast, T2DM led to a significant reduction in basal synaptic transmission in the hAPP NL/F mice (Fig 4A, right panel). On the other hand, paired-pulse facilitation (PPF), a form of short-term, activity-dependent synaptic plasticity common to most chemically transmitting synapses, was similar in WT and hAPP NL/F mice with T2DM (Fig 4B). Finally, T2DM did not alter long-term potentiation (LTP) neither in WT nor in hAPP NL/F mice (Fig S3A and B). Although learning processes are largely mediated by intense electrical activity sustained over time, low-rate activity at individual synapses (basal synaptic transmission) is essential for information processing (Panatier et al, 2011), leading us to assume that the behavioral deficits observed in hAPP NL/F mice are because of an effect of T2DM on the mechanisms underlying basal transmission.

### T2DM differentially affects dendritic spines in WT and hAPP NL/F mice

Dendritic spines are small protrusions of dendrites involved in processing stimuli arriving from excitatory terminals, fundamentally glutamatergic, as such constituting an important substrate in learning and memory processes (Zhou et al, 2004; Kasai et al, 2010). Therefore, we next went on to determine whether the changes in basal transmission produced by T2DM in hAPP NL/F mice had a correlation at the level of dendritic spines. The number and morphological characteristics of dendritic spines were assessed on brain sections of the same mice used for electrophysiology after local application of the well-characterized DiI dye (see the Materials and Methods section). Fig 5A shows representative confocal microscopy images of the appearance of these structures in the neurons of mice with different genetic backgrounds and experimental conditions, resulting in the apparent effect of T2DM, particularly in WT neurons. By quantifying labelled dendrites from neurons in different fields, a series of important data emerges (Fig 5B). One of them is the significant low number of dendritic spines in the cortical neurons from hAPP NL/F mice compared with WT. Another fact that emerges from this figure is the effect of T2DM, which significantly reduces the density of dendritic spines in WT mice, not in hAPP NL/F, although in this case, the lack of effect may be because of the fact that these mice already have fewer spines compared with the WT. In addition, the quantitative analysis

---

by Tukey's multiple comparisons test (*$P < 0.05$, ***$P < 0.001$). **(D, E, F)** Barne Maze. **(D)** The graph shows the primary latency during training days, WT and hAPP NL/F knockin (NL/F) mice, control (CTL), and T2DM (WT-CTL n = 11, WT-T2DM n = 14, NL/F-CTL n = 15, NL/F-T2DM n = 13). One-way ANOVA revealed no significant differences among experimental conditions. **(E)** Time mice spent exploring the target quadrant, where the escape hole is located, on test day (escape hole is blocked), WT and hAPP NL/F knockin (NL/F) mice, control (CTL), and T2DM (WT-CTL n = 11, WT-T2DM n = 14, NL/F-CTL n = 15, NL/F-T2DM n = 13). Statistical analysis was performed by one-way ANOVA ($P = 0.0044$), followed by Tukey's multiple comparisons test (*$P < 0.05$, ***$P < 0.001$, *n.s*, not significant). **(F)** The graph shows the percentage of mice using the different spatial strategies to locate the escape hole on test day (WT-CTL n = 11, WT-T2DM n = 14, NL/F-CTL n = 15, NL/F-T2DM n = 13). **(G)** The graph compares the spontaneous alternations in the Y-Maze paradigm, between WT and hAPP NL/F knockin (NL/F) mice, control (CTL), and T2DM (WT-CTL n = 7, WT-T2DM n = 7, NL/F-CTL n = 6, NL/F-T2DM n = 6). Statistical analysis was performed by Kruskal–Wallis test ($P = 0.0156$), followed by Dunn's multiple comparisons test (**$P < 0.01$). **(F)** Mann–Whitney test also revealed a significant effect of T2DM in NL/F mice ($^{\#}P = 0.0152$) All plots (except (F)) show mean values ± SEM.

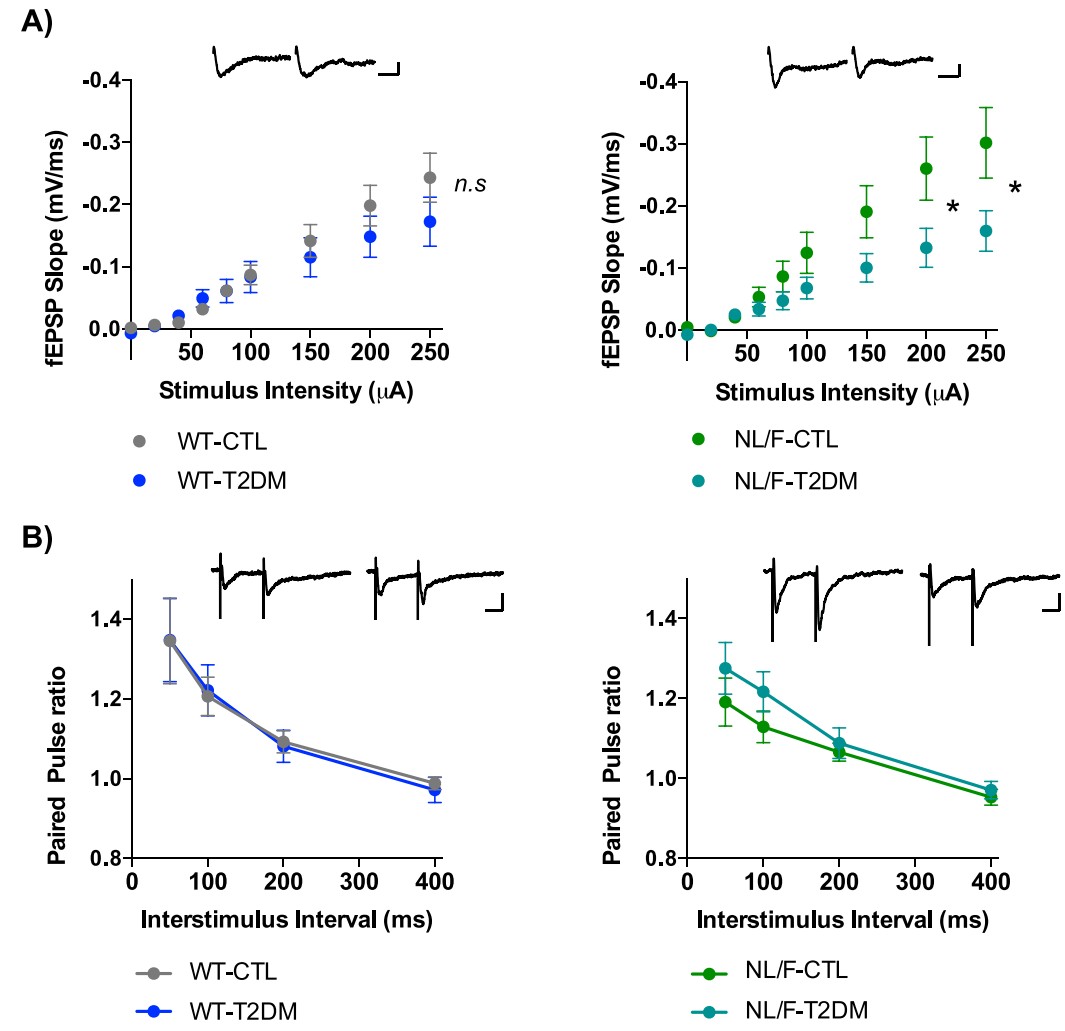

**Figure 4. Synaptic transmission is affected by T2DM in hAPP NL/F mice.**
**(A)** Basal synaptic transmission in WT (left panel) and hAPP knockin NL/F (NL/F) mice (right panel), control (CTL) or T2DM. (WT-CT n = 12, N = 7; WT-T2DM n = 11, N = 6; NL/F-CTL n = 14, N = 7; NL/F-T2DM n = 11, N = 5). Statistical analysis was performed by Mann–Whitney test (*n.s*, not significant, *P = 0.0108 [200 μA], *P = 0.0246 [250 μA]). Representative traces of responses to 250 μA stimulation are included (left trace CTL, right trace T2DM) for both genetic backgrounds. Scale bar: −0.1 mv/10 ms. **(B)** Paired Pulse Facilitation ratio in WT (left panel) and hAPP knockin NL/F (NL/F) mice (right panel), CTL or T2DM (WT-CTL n = 10, N = 7; WT-T2DM n = 10, N = 5; NL/F-CTL n = 10, N = 6; NL/F-T2DM n = 10, N = 5). Mann–Whitney test revealed no significant differences induced by T2DM. Representative traces of responses to paired pulse facilitation protocol, 50 ms interstimulus interval, are included (left trace CTL, right trace T2DM) for both genetic backgrounds. Scale bar: −0.1 mV/20 ms. All plots show mean values ± SEM. n, number of slices; N, number of animals.

revealed that T2DM induced a significant reduction in the proportion of mushroom-shaped spines in neurons from hAPP NL/F mice compared with WT (Fig 5C). Furthermore, T2DM led to an increase in the proportion of thin-shaped spines in hAPP NL/F mice with T2DM, not in WT (Fig 5D). The reduction of mushroom-shaped spines in hAPP NL/F mice with T2DM and the increase in thin spine could explain, at least in part, the deficits in basal transmission of these mice. In fact, a number of observations indicated that spine head volume (mushroom spines) positively correlates with synaptic strength (Matsuzaki et al, 2001). Consistently, the increase in stubby-shaped spines in WT mice after T2DM (Fig S4) could explain why these mice do not have an electrophysiological phenotype despite presenting a reduction in the total number of spines.

## T2DM differentially affects γ-secretase activity in WT and hAPP NL/F mice

Intuitively, the more profound behavior, electrophysiology, and architectural changes that occur in the hAPP NL/F mice when affected by T2DM would be the elevated levels of Aβ42 peptide that these mice produce from an early age (see Introduction). In fact, amyloid beta oligomers have been shown to be toxic to brain cells by different mechanisms, including dendritic spine loss (Manczak et al, 2018; Reiss et al, 2018). We therefore carried out ELISA assays to quantify the levels of Aβ peptide, both in the soluble and insoluble forms in the cortex of WT and hAPP NL/F mice with and without T2DM. Soluble and insoluble Aβ40 and Aβ42 levels were not modified by T2DM in the cortex of WT animals (Figs 6A and B and S5A and B). On the contrary,

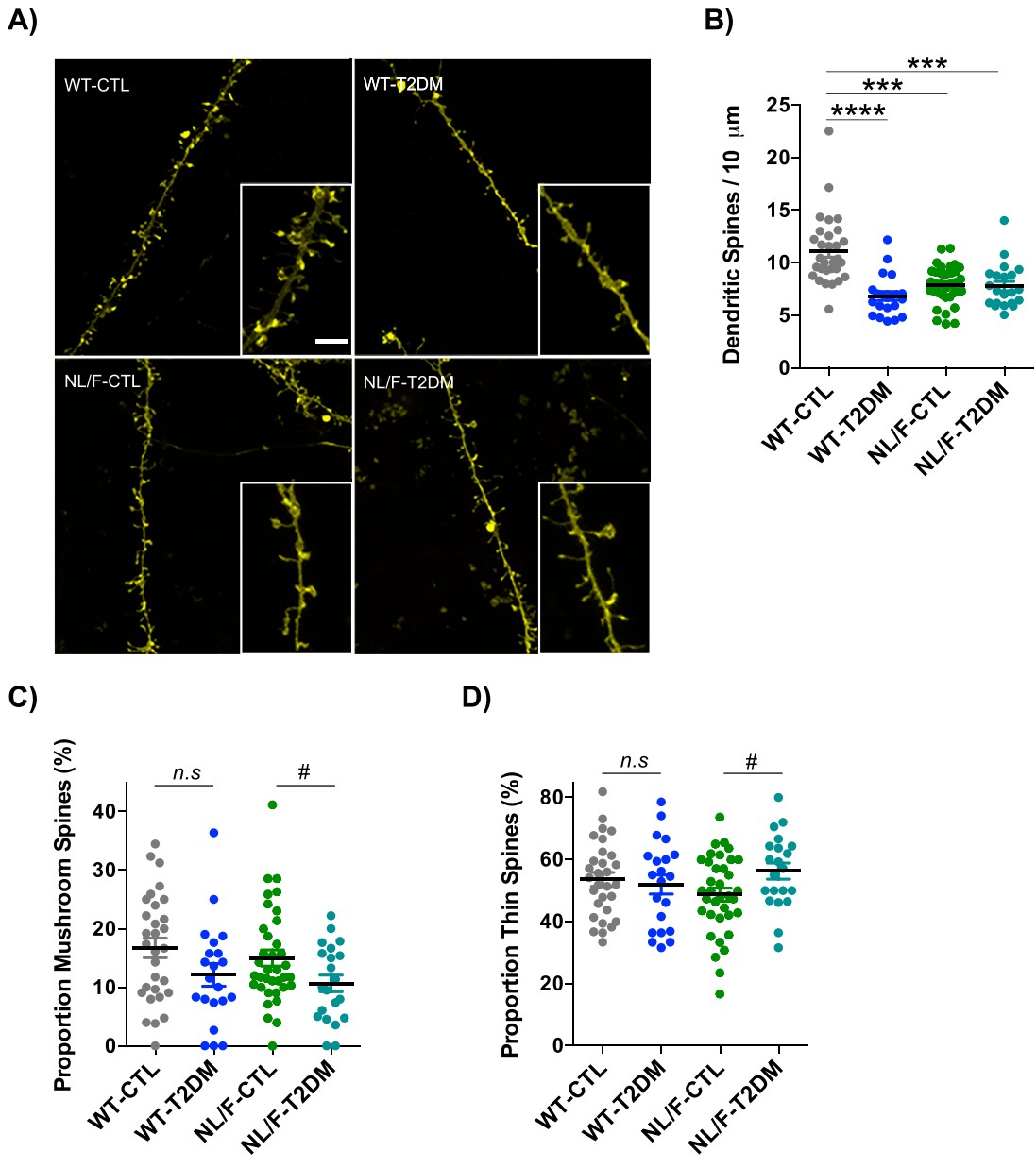

**Figure 5.  T2DM affects dendritic spine number and morphology.**
**(A)** Representative confocal images and their regions of interest magnified at their respective bottom right cornesr. Scale bar: 5 μm. **(B)** The plot compares the differences in the dendritic spines number in control (CTL) or T2DM-induced (T2DM) WT and hAPP knockin NL/F (NL/F) mice, (WT-CTL n = 30, WT-T2DM n = 21, NL/F-CTL n = 36, NL/F-T2DM n = 21). Statistical analysis was performed by Kruskal–Wallis test ($P < 0.0001$), followed by Dunn's multiple comparisons test (***$P < 0.001$, ****$P < 0.0001$). **(C)** Proportion of mushroom-shaped spines, expressed as percentage (WT-CTL n = 30, WT-T2DM n = 21, NL/F-CTL n = 36, NL/F-T2DM n = 21). Statistical analysis was performed by one-way ANOVA ($P = 0.0464$), post hoc analysis with Tukey's multiple comparisons test revealed no significant differences among experimental conditions. Unpaired $t$ test revealed a significant effect of T2DM in NL/F mice, but not in WT (n.s, not significant, #$P = 0.0436$). **(D)** Proportion of thin-shaped spines, expressed as percentage (WT-CTL n = 30, WT-T2DM n = 21, NL/F-CTL n = 36, NL/F-T2DM n = 21). One-way ANOVA revealed no significant differences among experimental conditions. Unpaired $t$ test revealed a significant effect of T2DM in NL/F mice, but not in WT (n.s, not significant, #$P = 0.0297$). The Plots show mean values ± SEM. Individual data points represent the number of dendrites in which spines were analyzed. Density and morphometric variables were quantified in three different dendrites from three different brain slices for each individual (three to four individuals per experimental condition).

T2DM led to a significant reduction of Aβ42 peptide in the hAPP NL/F mice, both the soluble and insoluble forms (Fig 6A and B). The levels of Aβ40 were not significantly affected by T2DM in these mice, probably because of the fact that these mice are fundamentally producers of the Aβ42 form (Saito et al, 2014). Consistent with this observation, T2DM resulted in a reduction (insoluble fraction) of the Aβ42/Aβ40 ratio in these mice (Fig S5C and D). Furthermore, hAPP NL/F mice with T2DM appeared to have fewer plaques than non-diabetic mice (Fig S6A and B). In addition, plaques in hAPP NL/F mice with T2DM were smaller (Fig S6C).

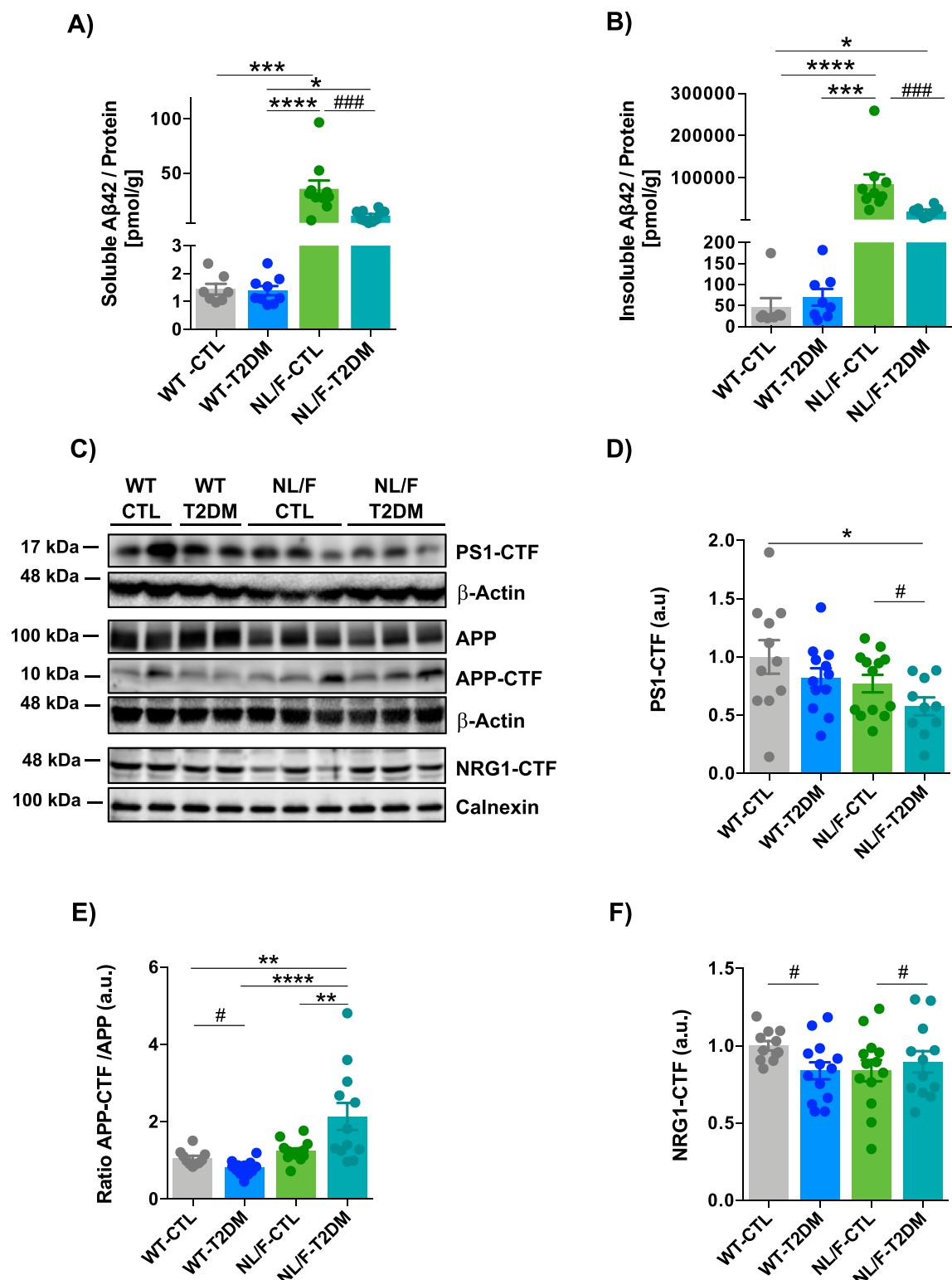

**Figure 6. T2DM affects γ-secretase activity differently in WT and hAPP NL/F mice.**
**(A)** Plot showing the comparison of the Aβ42 peptide content in the homogenization buffer (HB)-soluble fraction extracted from cerebral cortices of WT and hAPP NL/F knockin (NL/F) mice with T2DM or control condition (CTL), (WT-CTL n = 7; WT-T2DM n = 9; NL/F-CTL n = 10; NL/F-T2DM n = 10). Statistical analysis was performed by Kruskal–Wallis test (P < 0.0001), followed by Dunn's multiple comparisons test (*P < 0.05, ***P < 0.001, ****P < 0.0001). Mann–Whitney test also revealed a significant effect of T2DM in NL/F mice (###P = 0.0005). Aβ42 content was normalized to the total protein present in this fraction and determined by BCA assay. **(B)** Plot showing the comparison of the Aβ42 peptide content in the guanidine–hydrochloride-soluble fraction (Insoluble Aβ) extracted from cerebral cortices of WT and NL/F mice, CTL and

A possible explanation to the above results is that T2DM has an inhibitory effect on the activity of the enzymatic γ-secretase complex when its activity is enhanced towards the Aβ48–Aβ45–Aβ42–Aβ38 production line (Zhou et al, 2019), as is the case in the hAPP NL/F mice. To address this possibility, we analyzed the levels of the C-terminus form of presenilin 1 (PS1-CTF), the catalytic component of the γ-secretase complex responsible for the Aβ peptide generation after cleavage at the β site. Fig 6C and D show that T2DM was paralleled by a significant reduction in PS1-CTF levels in the cerebral cortex of hAPP NL/F mice, not in WT with diabetes. To complement these data, we then analyzed the levels of APP-CTF, the substrate for Aβ production that results of β-secretase-mediated cleavage. This experiment revealed higher levels of APP-CTF in the cerebral cortex of hAPP NL/F mice with T2DM compared with non-diabetic mice (Fig 6C and E). On the contrary, T2DM decreased the levels of APP-CTF in WT mice with T2DM (Fig 6C and E). These last results suggest that T2DM decreases γ-secretase activity in this particular, AD-prone, genetic background. To ascertain whether these effects were specific for APP processing, we next analysed the cleavage of Neuregulin 1 (NRG1), a key factor for cell communication during developmental processes and in the adult brain. Full-length NRG1 undergoes proteolytic cleavage by the β-cleavage enzyme BACE1 or by ADAM17 resulting in paracrine or juxtacrine signalling. The remaining C-terminus fragment (NRG1-CTF) is then cleaved by the γ-secretase complex in the transmembrane domain to generate an intracellular domain (Willem, 2016). Analysis of NRG1-CTF levels in the cortex of WT and hAPP NL/F revealed significantly higher levels in hAPP NL/F mice with T2DM compared with WT with T2DM (Fig 6C and F), reinforcing the notion that T2DM reduces γ-secretase activity for multiple substrates, but only in mice with a genetic background of predisposition to AD.

### T2DM reduces the levels of synaptic proteins in hAPP NL/F mice, not in WT

Previous works have shown that inhibition of γ-secretase by bath-application of DAPT or Compound E causes a reduction of synaptic potentiation and inhibits LTP (Chen & Behnisch, 2013), inhibits vesicle endocytosis (Sha et al, 2014), and genetic deletion of Presenilin 1 impairs synaptic facilitation (Barthet et al, 2018). Furthermore, previous work demonstrated that γ-secretase activity can take place at synapses where it regulates dendritic spine formation (Inoue et al, 2009). Therefore, the

observed inhibitory effect of T2DM on γ-secretase activity moved us to analyze the effect of this metabolic condition on proteins for synaptic function, focusing on PSD95, which is the core protein of the postsynaptic density and directly anchors neurotransmitter receptors at the synapse (Keith & El-Husseini, 2008). Moreover, recent work demonstrated that PSD95 protects synapses from the toxic effect of β-amyloid (Dore et al, 2021). Fig 7A and B reveal two important aspects of PSD95 levels in these mice: (i) reduced levels in the hAPP NL/F mice compared with WT in the control diet situation, and (ii) significant reduction by T2DM in hAPP NL/F mice, not in WT. To determine whether the last effect was related to an inhibitory action of T2DM on γ-secretase complex activity, we analyzed PSD95 levels in extracts of primary cortical neurons in vitro treated with DAPT, a chemical inhibitor of γ-secretase activity (Dovey et al, 2001). Immunoblot analysis of 15 d in vitro (15 DIV) mouse cortical neurons treated with DAPT for 24 h revealed a significant reduction in PSD95 levels (Fig 7C). Similar reduction was observed by immunofluorescence microscopy (Fig 7D and E). Because PSD95 promoter activity is regulated by the γ-secretase cleaved form of NRG1 (Bao et al, 2004), this reduction in PSD95 expression upon γ-secretase inhibition comes in further support of the negative effect that T2DM exerts on γ-secretase activity. Given that postsynaptic proteins such as PSD95 are key determinants for synapse structure and strength (Meyer et al, 2014), reduced activity of γ-secretase could be contributing to the reduced synaptic transmission and cognitive defects that T2DM produces in individuals with a genetic predisposition to AD.

### T2DM increases tau phosphorylation in the cerebral cortex of hAPP NL/F mice, not in WT

In addition to affecting postsynaptic transmission by inhibiting the γ-secretase complex, T2DM could affect transmission by affecting the level of phosphorylation of the microtubule-associated protein tau. In fact, previous work has demonstrated that the aberrant phosphorylation of tau in Serine epitopes 396 or 404 leads to its localization to dendritic spines which then produces postsynaptic dysfunction (Teravskis et al, 2021). It was also shown that phosphorylated forms of tau induce the loss of dendritic spines (Kandimalla et al, 2018). Therefore, we next analyzed the levels of phosphorylation of tau in the epitopes Serine 396 and Serine 404. Fig 8A and B show that T2DM did not affect the mean level of phosphorylation of tau in Serine 404 in the cerebral

T2DM (WT-CTL n = 7; WT-T2DM n = 8; NL/F-CTL n = 9; NL/F-T2DM n = 8). Statistical analysis was performed by Kruskal–Wallis test ($P < 0.0001$), followed by Dunn's multiple comparisons test (*$P < 0.05$, ***$P < 0.001$, ****$P < 0.0001$). Mann–Whitney test also revealed a significant effect of T2DM in NL/F mice (####$P = 0.0006$). **(C)** Representative immunoblots of PS1-CTF, APP, APP-CTF, and NRG1-CTF in total lysates of brain cortical samples from CTL or T2DM WT and NL/F mice. **(C, D)** Graph showing relative PS1-CTF protein levels (quantified from immunoblot experiments as the one shown in panel (C)) normalized to the WT control group (WT-CTL n = 11; WT-T2DM n = 12; NL/F-CTL n = 13; NL/F-T2DM n = 10). Statistical analysis was performed by one-way ANOVA ($P = 0.0471$), followed by Tukey's multiple comparisons test (*$P < 0.05$). Unpaired t test also revealed a significant effect of T2DM in NL/F mice (#$P = 0.0163$, data normalized to the NL/F control group). Actin was used as loading control. **(C, E)** Graph showing the ratio of APP-CTF/APP protein levels (quantified from immunoblot experiments as the one shown in panel (C)) normalized to the WT control group (WT-CTL n = 10; WT-T2DM n = 12; NL/F-CTL n = 13; NL/F-T2DM n = 12). Statistical analysis was performed by one-way ANOVA ($P < 0.0001$), followed by Tukey's multiple comparisons test (**$P < 0.01$, ****$P < 0.0001$). Unpaired t test also revealed a significant effect of T2DM in WT mice (#$P = 0.0081$). Actin was used as loading control. **(C, F)** Graph showing relative NRG1-CTF protein levels (quantified from immunoblot experiments as the one shown in panel (C)) normalized to the WT control group (WT-CTL n = 11; WT-T2DM n = 13; NL/F-CTL n = 13; NL/F-T2DM n = 12). One-way ANOVA revealed no significant differences among experimental conditions. Unpaired t test revealed a significant decrease mediated by T2DM in WT mice (#$P = 0.0229$), and a significant increase in NL/F mice (#$P = 0.0273$, data normalized to the NL/F control group). Calnexin was used as loading control. All plots show mean values ± SEM.

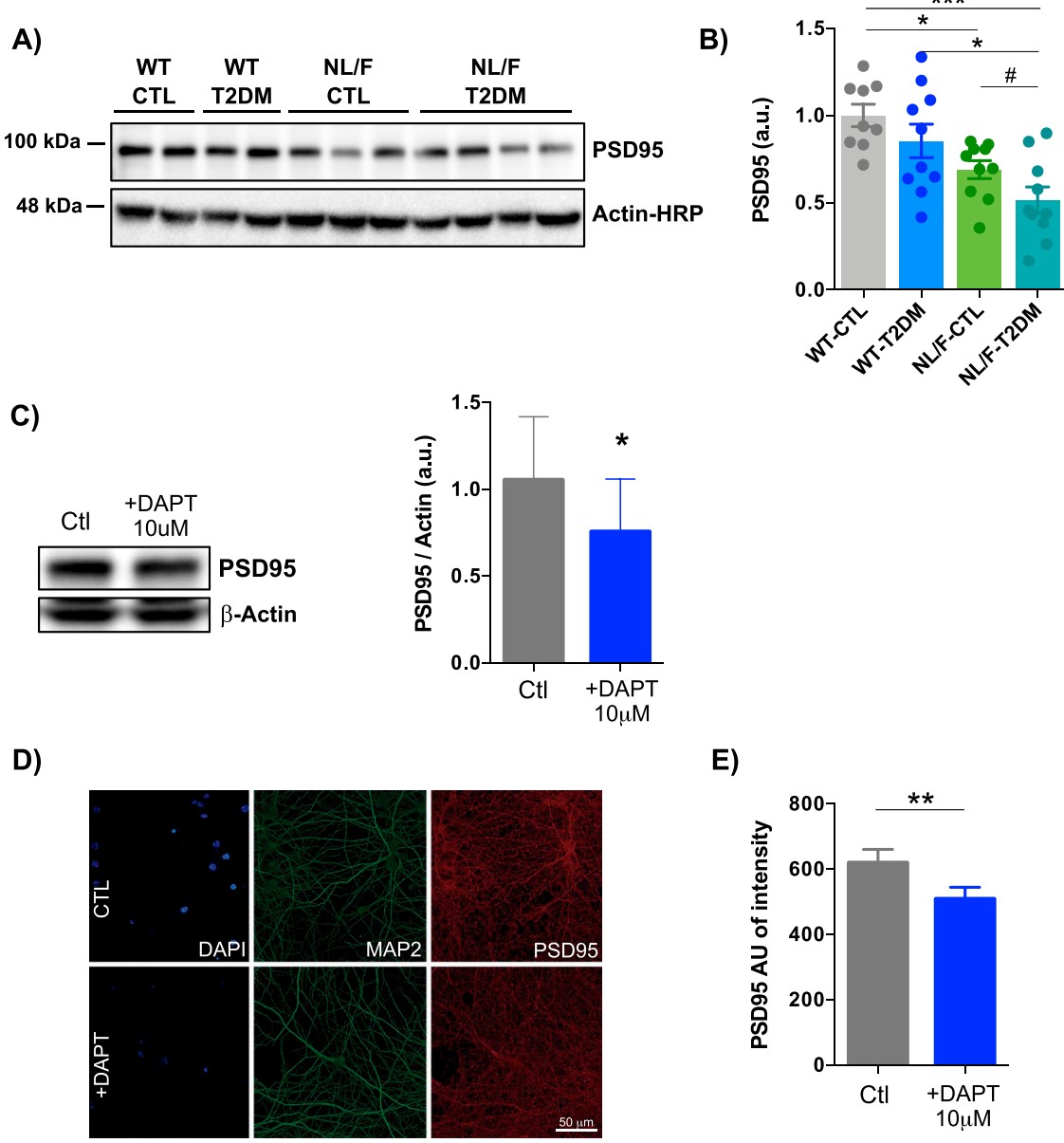

**Figure 7.   T2DM reduces PSD95 levels in hAPP NL/F but not in WT mice.**
**(A)** Representative immunoblots of PSD95 in total lysates of brain cortical samples from control (CTL) or T2DM WT or hAPP-NL/F knockin (NL/F) mice. Actin was used as loading control. **(A, B)** The graph shows PSD95 protein levels (quantified from experiments as the one shown in panel (A)) normalized to the WT control group, (WT-CTL n = 9, WT-T2DM n = 10, NL/F-CTL n = 10, NL/F-T2DM n = 10). Statistical analysis was performed by one-way ANOVA ($P = 0.0004$), followed by Tukey's multiple comparisons test (*$P < 0.05$, ***$P < 0.001$). Unpaired $t$ test also revealed a significant effect of T2DM in NL/F mice ([#]$P = 0.0271$, data normalized to NL/F control group). **(C)** DAPT treatment for 24 h in primary cortical neurons significantly decreased PSD95 protein levels determined by immunoblot (left). A plot comparing the average levels of PSD95 between control neurons (Ctl) and neurons treated with 10 $\mu$M DAPT (+DAPT), quantified from immunoblot experiments as the one shown in this panel, is shown on the right side of this panel (Ctl n = 6, DAPT n = 6). Statistical analysis was performed by paired $t$ test (*$P = 0.0461$). **(D)** Representative confocal images for DAPI, MAP2, and PSD95 in control conditions (upper images) and treated with DAPT inhibitor (lower images). Scale bar: 50 $\mu$m. **(D, E)** The plot compares the average immunofluorescence intensity values for PSD95, quantified from confocal images as the ones shown in panel (D) (Ctl n = 5, DAPT n = 5). Statistical analysis was performed by paired $t$ test (**$P = 0.0083$). All plots show mean values ± SEM.

cortex of WT animals but did significantly increase it in the cerebral cortex of hAPP NL/F mice. An identical conclusion can be reached by observing the data in Fig 8A and C, which reflect the effect of T2DM on the levels of phosphorylation of tau in the epitope Serine 396: lack of effect in WT mice and significant increase in hAPP NL/F mice.

## Comparative analysis of the transcriptome of the cerebral cortex of WT and hAPP NL/F mice: significant differences in membrane transport/import pathways (and in thermoregulation)

To define how the genetic background might have played a part in the increased vulnerability of hAPP NL/F mice to T2DM, we

A)

WT CTL | WT T2DM | NL/F CTL | NL/F T2DM

63 kDa — Tau5
100 kDa — Calnexin
63 kDa — Tau5-P S404
100 kDa — Calnexin
63 kDa — Tau5-P S396
100 kDa — Calnexin

**Figure 8. T2DM increased tau phosphorylation in hAPP NL/F but not in WT mice.**
**(A)** Three representative immunoblots of total tau, tau Ser 404 phosphorylation, and tau Ser 396 phosphorylation in total lysates of brain cortical samples obtained from control (CTL) or T2DM-induced (T2DM) WT and hAPP knockin NL/F (NL/F) mice. Calnexin was used as loading control. **(B)** The graph shows tau Ser 404 phosphorylation protein levels determined by immunoblot (WT-CTL n = 11, WT-T2DM n = 13, NL/F-CTL n = 13, NL/F-T2DM n = 12). Statistical analysis was performed by one-way ANOVA ($P = 0.0281$), followed by Tukey's multiple comparisons test (*$P < 0.05$). **(C)** The graph shows tau Ser 396 phosphorylation protein levels (WT-CTL n = 11, WT-T2DM n = 13, NL/F-CTL n = 13, NL/F-T2DM n = 13). Statistical analysis was performed by one-way ANOVA ($P = 0.0297$), followed by Tukey's multiple comparisons test (*$P < 0.05$). Plots show relative protein levels normalized to total tau and to the WT control group. All plots show mean values ± SEM.

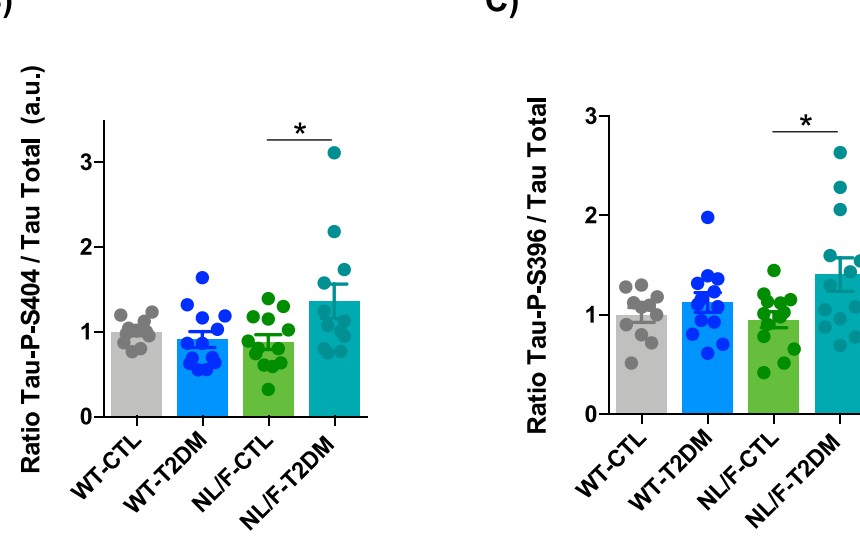

B)

C)

performed RNA sequencing of the cerebral cortex of 15-mo-old hAPP NL/F and WT under standard diet conditions. Fig 9A shows that the cerebral cortex of hAPP NL/F mice differs from that of WT mice in the expression levels of about 50 transcripts, of which 1/3 are up-regulated and 2/3 down-regulated. Fig 9B is the functional association of the differentially expressed genes (DEGs) by means of an over-representation analysis (ORA), a widely used approach to determine whether known biological functions or processes are overrepresented (i.e., enriched) in an experimentally derived gene list (Table S1). In Fig 9B, it can be clearly seen that the enrichment of different functional pathways, the most represented being those of genes involved in membrane transport (of lipids, sugars, ions) and in temperature homeostasis. Although it is relatively simple to explain how changes in the levels of transmembrane channels could increase the vulnerability of these mice to the effects of T2DM, this association is not so simple in view of the changes in genes involved in thermoregulation. One possibility, however, would be

that, perhaps because of the effect of amyloid oligomers on the plasma membrane of thermoregulatory neurons in the hypothalamus, changes in body temperature were produced, which have been seen to lead to changes in tau phosphorylation (Vandal et al, 2016; Tournissac et al, 2021).

## Discussion

The main message of this work is that T2DM has a greater impact on cognitive function when there is a background of brain pathology/pre-pathology. In fact, mice with the hAPP NL/F background developed larger deficits on cognitive function and synaptic transmission when exposed to a T2DM protocol. However, even in these mutant mice, T2DM did not induce symptoms and signs typical of AD, only cognitive and biochemical alterations more intense than in

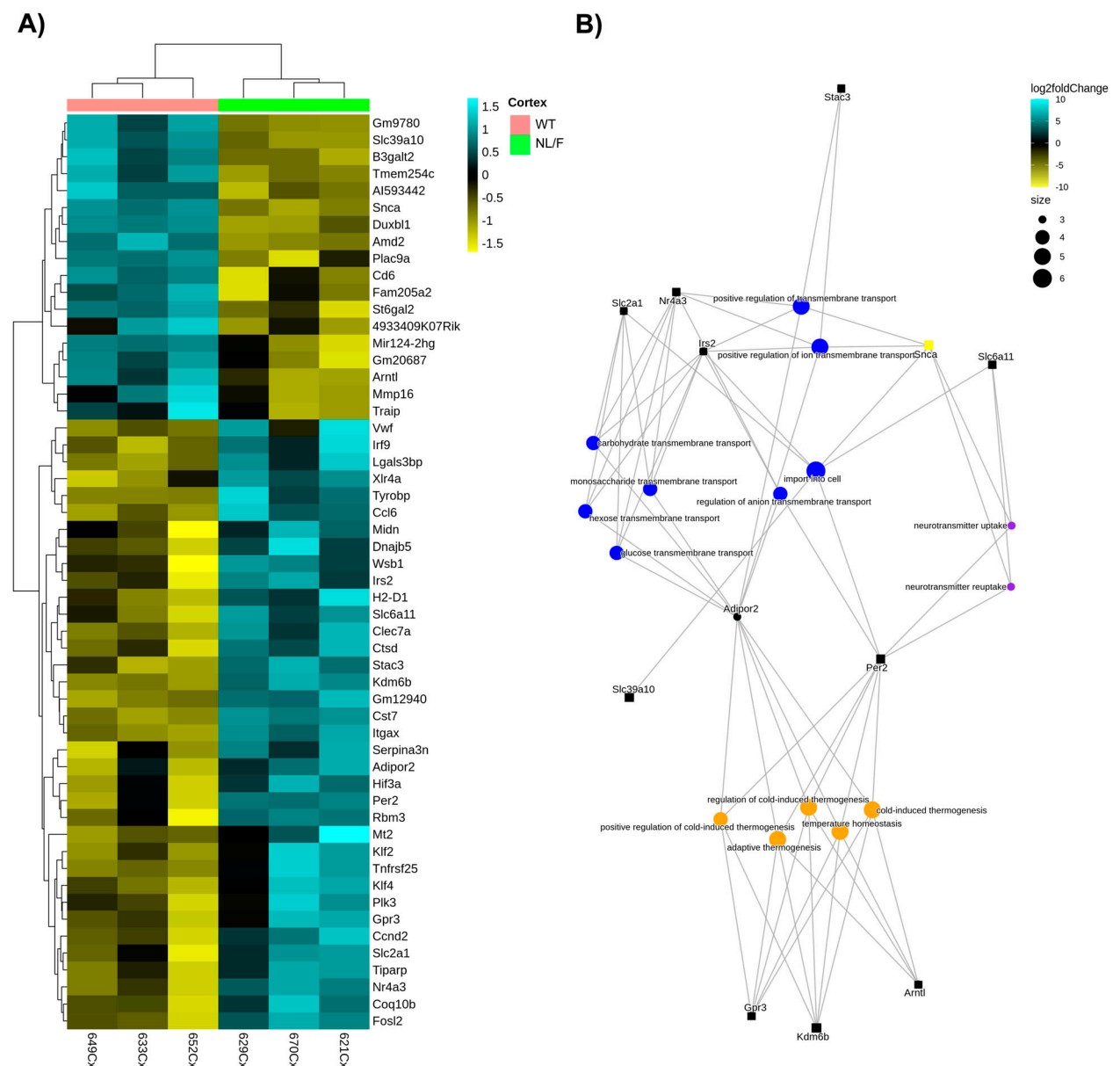

**Figure 9. Representation of Differential Gene Expression (DEGs) with q-value < 0.05 identified in the cortex between the NL/F standard diet (NL/F) and the wild-type standard diet (WT) groups.**

**(A)** Heatmap comparison by hierarchical cluster analysis. Each row represents a single gene and each column represents a condition sample. Gene intensities are $\log_2$ transformed and displayed as colors ranging from yellow to blue representing the changing process from down-regulation to up-regulation as shown in the key. **(B)** Gene-concept network (cnetplot) of overlapping DEGs set. The graph shows the enriched GO terms (concepts) obtained with an FDR cut-off of 0.05 linked to the DEGs involved in a process as a network. For simplification, only 15 of the top 20 enriched biological processes related to transmembrane transport (blue dots), thermogenesis (orange dots), and neurotransmission (purple dots) are displayed. The size of dots (concept nodes) depends on the gene count involved in that process. The color of the gene nodes (squares) depends on their $\log_2$ fold change values according to the displayed color gradient ranging from yellow to blue.

animals of the same age and the same genetic background that did not have diabetes. The lack of a severe phenotype by T2DM condition in our mice is possibly not too far from what happens in humans, in whom, only modest impairments in complex attention, information processing, and executive function occur in individuals with long-term T2DM (Arnold et al, 2018). These results are in contrast with other mouse models of AD, where a more dramatic worsening of AD signs and symptoms by T2DM was observed

(reviewed in Salas & De Strooper [2019]). The explanation that we favor is based on the different levels of brain damage of the mice at the time diabetes is induced. In the hAPP NL/F mice used in this study, the increase in Aβ is the consequence of the knock-in of a humanized version of the mouse gene, that is, replacing the endogenous APP in its genetic locus. Therefore, the expression levels of APP are the consequence of the activity of the intrinsic promoter, and only in the cells in which the gene is physiologically expressed.

In contrast, the results where obvious signs of AD by T2DM have been reported are based on mutant APP and presenilin in transgenic conditions, therefore, expressing the mutant proteins in cells that would not necessarily express them and at levels that are not regulated by the endogenous promoter (Kalback et al, 2002; Saito et al, 2016). Without ruling out the biological significance of the results in transgenic mice, the results obtained in the knockin hAPP NL/F mice used here seem closer to what occurs in the most common form of human T2DM: that is, adult-onset without concomitant—evident—brain pathology.

There are many possible explanations as to why mice with the hAPP NL/F background are more susceptible to showing defects in the presence of T2DM than WT. One possibility is that exposure to elevated levels of Aβ from an early age in these mice has provoked biochemical alterations that lower the threshold for the damaging action of T2DM associated to inflammation. Although previous data would indicate that this is a real possibility, mice with T2DM, whether in the WT or hAPP NL/F genetic background, did not show major signs of inflammation, at least judging by the number and morphology of microglial cells (Fig S7). On the other hand, the high production of Aβ42 peptide from early ages in these mice could, via oligomers, produce changes in the plasma membrane leading to ionic imbalance and consequently to functional defects at multiple levels (Benilova et al, 2012; Mucke & Selkoe, 2012; Smith & Strittmatter, 2017; Huang & Liu, 2020). Accordingly, our bioinformatics analysis confirmed that a primary alteration in the hAPP NL/F mice is at the level of plasma membrane transporters of lipids, sugars, and ions. These findings are in agreement with previous reports showing that the levels of several glucose transporters are altered in the brain of AD patients (Kyrtata et al, 2021). In addition, the first GWAS studies conducted on large cohorts of sporadic AD patients supported the notion that processes involving membrane transport were involved in AD pathology, through the identification of AD-associated SNPs in the ATP-binding cassette transporter, members 7 and 1 (ABCA7 and ABCA1, respectively) genomic region (Bellenguez et al, 2022), two proteins involved in the transport of several substrates across cellular membranes (Dean & Annilo, 2005). Although the changes in membrane permeability could have relevance in the phenotype induced by T2DM in a background of predisposition to AD, we do not rule out that the changes that occur in other genes also contribute to the final phenotype.

The next aspect to consider is the mechanism behind the effect of T2DM. Clearly, what T2DM does to the mice should be similar in the two genetic backgrounds, simply that the effect at the brain level is greater in the hAPP NL/F mice, conceivably because of some of the weaknesses that occur in these mice. One mechanism by which T2DM could lead to alterations is hypoxia. In fact, hypoxia regulates γ-secretase activity and proteolytic γ-secretase function is needed for proper activation of the response to hypoxia (Kaufmann et al, 2013; Gertsik et al, 2014). Given that γ-secretase is an important regulator of synaptic function (Chen & Behnisch, 2013; Barthet et al, 2018), it does not seem unreasonable to propose that some of the cognitive defects in hAPP NL/F mice with T2DM arise as consequence of reduced γ-secretase activity. This scenario seems counterintuitive, because γ-secretase activity is also required for amyloid peptide production, a pathogenic element in AD. However, the interpretation that γ-secretase activity is required to preserve cognitive function does appear to apply also in humans, because two large phase 3 clinical trials with the γ-secretase inhibitor Semagacestat were prematurely interrupted because of the observation of detrimental effects on cognition and functionality (Doody et al, 2013). Another mechanism, linked to or independent from reduced γ-secretase activity, that could contribute to the cognitive deficits in animals with T2DM and a background of predisposition to AD is the increase in phosphorylation of tau in the Ser396 and Ser404 epitopes. Mutational studies with human tau indicate that phosphorylation at serine 396 and serine 404 leads to the functional loss of tau-mediated tubulin polymerization, dissociation of tau from microtubules, and miss-localization to dendrites (Evans et al, 2000; Teravskis et al, 2021). This phosphorylation-dependent tau dysfunction is considered one of the critical events leading to neuronal dysfunction and eventual degeneration (Arriagada et al, 1992; Avila et al, 2004; Hoover et al, 2010; Torres et al, 2021). A recent study showing that in vivo inactivation of presenilin accelerates tau phosphorylation and aggregation (Soto-Faguás et al, 2021) would indicate that abnormal tau phosphorylation in the hAPP NL/F and T2DM mice could downstream the inhibition of γ-secretase activity. Another mechanism behind the effect of T2DM in the hAPP NL/F mice is through the decrease in the expression of synaptophysin (Fig S8) which, added to the decrease in PSD95, would compromise function both at the pre and post-synaptic. In light of these results, it is tempting to propose that cognitive defects in the hAPP NL/F mice with T2DM are the result of alterations at, at least, two different levels: directly dependent on the perturbations induced by amyloid peptide/oligomers over time (i.e., intracellular ionic imbalance) and directly dependent on the alteration induced by a particular external insult (i.e., γ-secretase inhibition/Tau aberrant phosphorylation in the T2DM condition). Although our results suggest these mechanisms, more work is needed to define them precisely.

Although we have repeatedly stressed that the cognitive alterations by T2DM are the consequence of a two-hit process, T2DM/predisposition to AD, it is most likely that predisposition to AD is not the only background of brain disorder that would facilitate the development of cognitive alterations. In fact, it may very well happen that the development of cognitive deficits by T2DM also occur in individuals with another type of brain (or systemic) alteration, effective or latent. In fact, evidence exists that individuals with T2DM appear to be at increased risk of developing Parkinson's disease, and experiencing faster progression and a more severe phenotype of the disease (Cheong et al, 2020). Future work is needed to know if other conditions of brain vulnerability lead to behavioral and biochemical alterations similar or different to those reported here.

# Materials and Methods

### Animals and study design

The following animal procedures were reviewed and approved by the ethics committees of the Centro de Biología Molecular Severo Ochoa's (CBM) and Dirección General de Medio Ambiente de la

Comunidad Autónoma de Madrid (PROEX 204/19). All the animal procedures were performed in accordance with the guidelines of the European Union (2010/63/UE).

Adult WT and hAPP NL/F male and female mice (7–9 mo old) were used in this study. hAPP NL/F mice were made on C57BL/6J background as detailed in Saito et al (2014). All the animals were housed in the CBM animal facility in standard-sized cages in a ventilated rack, temperature- and humidity-controlled, on a 12-h light cycle (8 AM to 8 PM) with free access to food and water. The mice were randomly divided into two groups with standard rodent chow or a diet containing 60% kcal as fat (HFD) ad libitum. Body weight (BW) and basal glucose levels were measured every 2 wk. Mice received i.p. injections of vehicle (0.05 M citrate buffer, pH 4.5) or doses of STZ dissolved in the vehicle at 40 mg/kg BW per dose. STZ was prepared fresh every day and five doses were administrated, one dose per day on consecutive days. At 2–3 wk after the first injection, a GTT and ITT were performed. At weeks 3–4, the animals were transferred to the behavior room and/or were euthanized for electrophysiology, biochemical, and/or histological analysis.

## GTT and ITT

Basal glucose levels were measured in whole blood drawn from the tail vein using a hand-held glucometer (Accu Check Aviva). Both glucose and ITTs were performed after 6 and 4 h fasts, respectively, with baseline glucose measurement taken immediately before beginning the test. For GTT, glucose (Sigma-Aldrich) was filtered and administrated via i.p. injection at a dose of 0.2 g/kg BW in saline buffer 0.9%. Glucose levels were measured at 15, 30, 60, 90, and 120 min after injection. For ITT, human insulin (Actrapid) was delivered via i.p. injection at a dose of 0.8 UI/kg BW, and blood glucose measurements were taken at 15, 30, and 60 min after injection.

## Immunoblot

Cortex of WT or hAPP NL/F (control or T2DM) mice were dissected for biochemical studies and homogenized in RIPA buffer (20 mM Tris–HCl, pH 7.5, 150 mM NaCl, 1 mM EDTA, 1 mM EGTA, 1% NP–40, 1% sodium deoxycholate, 0.1% SDS) with protease inhibitors (cOmpleteTM, Sigma-Aldrich) and phosphatase inhibitors (Sigma-Aldrich). For primary cortical cultures, after the corresponding treatment, they were washed with cold PBS and lysed in RIPA buffer with protease and phosphatase inhibitors. The concentration of proteins present in the homogenized samples was determined using a PierceTM BCA Protein Assay Kit (Thermo Fisher Scientific), following the instructions of the commercial kit. Subsequent calculations were made to determine the appropriate volume of each homogenized sample so that all the immunoblot samples would have the same total protein amount (30 or 50 $\mu$g depending on the experiment). Proteins were prepared in Laemmli buffer (25 mM Tris–HCl pH 6.8, 1% SDS, 3.5% glycerol, 0.4% 2-mercaptoethanol and 0.04% bromophenol blue), and diluted in distilled water to reach a final volume of 30 $\mu$l. Individual animals' samples were run in each immunoblot experiment.

Proteins were separated by electrophoresis in polyacrylamide gels in the presence of SDS at constant voltage. Proteins were then transferred to nitrocellulose membranes by wet transfer, and after blocking with blocking solution (5% BSA in 0.1% Tween-20 in TBS [T-TBS]), membranes were incubated overnight with primary antibodies (diluted in blocking buffer) at 4°C with gentle agitation. This procedure was repeated three times for each animal sample. After washing the membranes with T-TBS, they were incubated with the corresponding secondary antibodies coupled to horseradish peroxidase and diluted 1:2,500 for 1 h at RT. Protein signal was detected with luminol (PierceTM ECL Western Blotting Substrate, Thermo Fisher Scientific) and chemiluminescence was measured using a CCD camera (Amersham Imager 680). To quantify differences in protein concentration, we carried out a linear range of exposure times of selected samples, for both target proteins and loading control proteins. Images were then analyzed to identify the protein band of interest (e.g., actin as loading control) in the linear dynamic range. Once the experimental setup and conditions were established, we repeated the conditions of the sample load, transfer method, transfer time, antibody dilution, antibody incubation time, and temperature in all subsequent experiments. The band corresponding to the protein of interest was quantified using FIJI software and normalized with respect to the values obtained for the loading control protein.

## Antibodies

The following antibodies were used for Immunoblot: mouse anti presenilin 1, dilution 1:1,000 (MAB5232; Merck Millipore); rabbit anti-Amyloid Precursor Protein, C-terminal antibody, dilution 1:1,000 (A8717; Sigma-Aldrich); rabbit anti-neuregulin 1, dilution 1:1,000 (NBP2-19588; Novus Biologicals); mouse anti-PSD95, dilution 1:1,000 (610495; BD Biosciences); mouse anti-tau5, dilution 1:1,000 (AHB0042; Thermo Fisher Scientific/Invitrogen); rabbit anti-tau-P Ser 396, dilution 1:1,000 (44752G; Life Technologies); rabbit anti-tau-P Ser 404, dilution 1:1,000 (44758G; Life Technologies); mouse anti-IRS1, dilution 1:1,000 (BD Biosciences); rabbit anti-IRS1-P Ser307, dilution 1:500 (ab1194; Abcam); rabbit anti-synaptophysin, dilution 1:1,000 (ab-14692; Abcam); rabbit anti-calnexin, dilution 1:10,000 (ab22595; Abcam); mouse anti-$\beta$-actin, dilution 1:10,000 (A5441; Sigma-Aldrich).

## Mouse primary cortical cultures

Cortical neurons were obtained from mouse embryos (E18), following a protocol adapted from previous descriptions (Kaech & Banker, 2006). Embryos were preserved in HBSS supplemented with 0.45% glucose and HEPES pH 7.3 and then cortices were dissected and preserved on ice. They were incubated in 0.0038% Trypsin–EDTA/HBSS (Gibco) for 15 min in an incubator. After tissue precipitation, HBSS was changed three times.The tissue was disintegrated and resuspended in plating medium (10% horse serum, 0.6% glucose, 2 mM glutamine in MEM) and lumps were removed with 70 $\mu$m filters. Neurons were seeded in coverslips previously treated with 0.5 mg/ml poly-L-Lysine (Sigma-Aldrich) for immunofluorescence experiments or on culture plates previously treated with 0.1 mg/ml poly-L-Lysine for biochemical studies. After 3 h, plating medium was exchanged to Neurobasal (Gibco) supplemented with GlutaMAX (Gibco) and B27 (Gibco). Neurons were placed in a humidified incubator at 37°C and 5% $CO_2$ for 1 wk. Then,

the medium was replaced with Neurobasal supplemented with B27. γ-secretase inhibitor DAPT (10 μM; D5942; Sigma-Aldrich) treatment was added during 24 h on DIV15 neurons.

## Behavioral tests

For behavioral experiments, mice from all experimental conditions were simultaneously transferred to a behavior room where they habituated for several days before the start of the tests. The light/dark cycle was 12/12 h (lights on 8:00 AM). All the experiments were performed during the light phase. To minimize variability, each animal was always tested at the same time of the day. The behavioral tests were carried out 18 wk after the beginning of the treatment (see Fig 1).

## Open Field and NOR tests

The Open Field (OF) and NOR were conducted in Open Field square test boxes (43 × 43 cm) over 3 d. The animals' behaviors were recorded with a video camera positioned above the box. During the first day, a habituation trial was conducted, in which the mice were freely allowed to explore the empty box for 8 min. In this day, the Open Field paradigm was evaluated and exploration and locomotor activity were assessed. The Open Field (OF) is based on the aversion of rodents to open spaces and their innate exploring instinct; they tend to move around the field. An increase in anxiety or a decrease in the willingness of the animal to move, affects the locomotion negatively. The results were analyzed using ANY-maze Video Tracking System software (StoeltingCo.).

24 h later, the NOR paradigm was performed. This task is based on the spontaneous tendency of rodents to spend more time exploring a novel object than a familiar one. The choice to explore the novel object reflects the use of learning and recognition memory. The mice were allowed to explore during 10 min in the same box, containing two equal objects. The following day (24 h later), one of the objects was replaced with a novel one, and the mice were allowed to freely explore the arena during 10 min. Placement of the novel object followed a counterbalanced design among trials to control for location effects.

For the behavioral scoring, object exploration was defined as the time the mice spent actively exploring the objects, that is, the mouse's head was within 2 cm of the object and directed towards it. The time the mice spent sitting on the objects without actively sniffing them did not account as object interaction.

The novel object preference index was calculated with the following formula: 100*time exploring novel object/(time exploring novel object+time exploring familiar object).

## Barnes Maze Test

Spatial learning and memory were assessed using the Barnes Maze Test, with a protocol adapted from Bárez-López et al (2017). The test was conducted on a Barnes Maze apparatus that consisted of an elevated circular platform of 1.2 m diameter with 20 equidistant holes of 45 mm diameter along the perimeter. One hole contained a dark box underneath (the target hole) that allowed the mice to escape the maze. Spatial extra-maze cues were positioned on the

curtain backdrop surrounding the maze at heights visible from the platform surface to allow the mice to orientate and locate the escape hole. These consisted of different colorful geometric shapes made of rigid paper, as described in Bárez-López et al (2017) and Gawel et al (2019). Because of the age of the animals, the mice were trained for four consecutive days so they could learn the position of the escape hole, and then on the fifth day, the memory test was performed. On the first day, a pretraining trial was conducted: the mice were placed in the center of the board inside an opaque container, the container was then lifted, and the mice were pretrained to enter the escape box by guiding them to the escape box and remaining inside for 1 min. After this, the first training trial started. Training sessions consisted of four trials per day: two consecutive trials followed by a 40 min rest and then another two consecutive trials. At the beginning of each trial, the mice were placed in the same opaque container in the center of the maze, the container was lifted, and the mice were allowed to freely explore the platform during 5 min or until they entered the escape box. Because rodents are aversive to bright light and open spaces, they start looking in random directions and start exploring until they enter the scape hole, considering this the end of the trial. If the mice did not enter the escape box after 5 min, they were guided to the escape box and remained there for 1 min. The memory test trial was performed on the fifth day: the target hole was blocked so mice could not escape, and the mice were allowed to explore for 2 min. The apparatus was cleaned with 0.1% acetic acid between trials. All the experimental trials were recorded using a video camera placed above the maze, and analyzed using ANY-maze Video Tracking System software (StoeltingCo.). Because measures based on locating the escape hole, that is, visiting and exploring it for the first time (primary measures), have been reported to be more sensitive than measures based on entering the hole (total measures) (O'Leary & Brown, 2013), we have focused our analysis on the primary measures.

## Y maze spontaneous alternations

Spatial working memory was evaluated with the Y-Maze spontaneous alternation paradigm. The test was conducted on a Y-shaped maze with three opaque arms oriented 120° from each other, labelled as A, B, and C. Spontaneous alternations are assessed by allowing mice to explore all three arms of the maze freely, which is driven by the innate curiosity of rodents to explore previously unvisited areas. If the mouse under analysis has a good working memory, it will remember which of the arms it has explored most of the time or recently and will prefer to visit the least explored arm.

The mice were gently placed at the distal part of the A arm facing the center and were allowed to freely explore the maze with all three arms open for 5 min. An alternation was defined as entries into three different arms consecutively. The percentage of alternations was calculated as the number of alternations divided by the total possible alternations (total arm entries minus 2) and multiplied by 100.

## Electrophysiology

### Acute hippocampal slices

Animals were anesthetized and quickly decapitated. The brains were rapidly removed and placed in a $Ca^{2+}$-free ice-cold dissection solution (10 mM D-glucose, 4 mM KCl, 26 mM $NaHCO_3$, 233.7 mM

sucrose, 5 mM $MgCl_2$, 1:1,000 phenol red as a pH indicator) previously saturated with carbogen (5% $CO_2$, 95% $O_2$). Coronal slices (300 $\mu m$ thick) were obtained cutting the brain in the same solution with a vibratome (VT1200S; Leica). Slices were incubated in carbogen-gassed recovery solution (92 mM choline chloride, 2.5 mM KCl, 1.2 mM $NaH_2PO_4$, 30 mM $NaHCO_3$, 20 mM HEPES, 25 mM glucose, 5 mM sodium ascorbate, 2 mM thiourea, 3 mM sodium pyruvate, 10 mM $MgSO_4.7H_2O$, 0.5 mM $CaCl_2.2H_2O$) for 15 min at 32°C. After that time, the slices were maintained at 25°C in carbogen-gassed aCSF (119 mM NaCl, 2.5 mM KCl, 1 mM $NaH_2PO_4$, 11 mM glucose, 26 mM $NaHCO_3$, 1.2 mM $MgCl_2$, 2.5 mM $CaCl_2$).

### Field recordings

Field excitatory postsynaptic potentials (fEPSPs) were recorded from CA3 to CA1 synapses in the hippocampal slices. The recording chamber was constantly perfused with aSCF, which was continuously gassed with 5% $CO_2$ and 95% $O_2$ and its temperature was monitored and maintained at 25°C. For all experiments, aCSF was supplemented with 100 $\mu M$ picrotoxin to block GABAA receptors. Glass recording pipettes (0.5–1.5 MΩ) were filled with aCSF and placed in the stratum radiatum of CA1. The stimulation electrode was placed on the Schaffer collateral fibers, between CA3 and the glass pipette. Synaptic responses were evoked at 0.067 Hz with 50 $\mu s$ stimuli. To generate an input–output curve, fEPSPs were recorded at different stimulation intensities (20–250 $\mu A$). Stimulation intensity was adjusted to 30% of the maximum response for LTP and PPF. PPF experiments were performed with pairs of pulses at 50, 100, 200, and 400 ms inter-stimulus intervals. LTP was induced after 20 min of stable baseline with a theta-burst protocol composed of 10 trains of bursts (four pulses at 100 Hz with a 200 ms interval) and it was repeated for four cycles with 20 s inter-cycle interval. Analysis was performed with custom-made Excel (Microsoft) macros.

### Immunofluorescence of neurons in culture

15 mm coverslips with neurons were fixed in 4% PFA, permeabilized with 0.1% Triton X-100 and incubated overnight with primary antibodies diluted in 3% BSA/PBS at 4°C: (anti-MAP2 [1:2,000, 822501; BioLegend], anti-PSD95 [1:1,000, 75-028; Neuromab]). Next day coverslips were incubated for 1 h at RT with secondary antibodies (Alexa-488 $\alpha$-chicken [1:1,000, A-11039; Thermo Fisher Scientific], Alexa-555 $\alpha$-mouse [1:1,000, A-31570; Thermo Fisher Scientific]) and DAPI (1:5,000). Finally, coverslips were mounted with ProLong (Thermo Fisher Scientific) and imaged at confocal microscope Nikon A1R+ with 60×/1.4 Plan Apochromat. Fluorescence quantification was done with FIJI software. A 12% threshold was applied over the MAP2 signal to create a mask. This was used over PSD95 channel to quantify a specific signal within neurons.

### Immunofluorescence of brain tissue

The left hemisphere of the brain was fixed for 24 h in 4% PFA and washed with PBS. Brain hemispheres were embedded in a solution of 10% sucrose–4% agarose in 0.2 N phosphate buffer (PB). The resulting blocks were cut with a vibratome (LeicaVT1200S) obtaining 40 $\mu m$ thick sagittal sections. Brain sections were stored at –20°C in

a cryoprotective solution (40% glycerol, 30% ethylene glycol, 20% 0.2 N PB, and 10% distilled water). Brain slices were washed with TBS, and permeabilized and blocked with a blocking solution (2% BSA and 0.5% Triton X-100 in TBS) for 1 h at RT. The sections were then incubated overnight at 4°C with primary antibodies diluted in blocking buffer: (anti-GFAP [1:500, MAB3402; EMD Milipore Corp] and anti-Iba1 [1:500, 019-19741; Wako]). After 3 washes with TBS, brain sections were incubated for 1 h at RT with secondary antibodies conjugated with fluorophores (Alexa 488 D anti-mouse [1:500, Life Technologies] and Alexa 647 D anti-rabbit [1:500, Life Technologies]). They were then washed with TBS and nuclei were stained with DAPI (1:5,000 in TBS). The sections were mounted on slides using ProLong (Thermo Fisher Scientific). Images were acquired on a LSM710 inverted confocal fluorescence microscope (ZEISS) with an oil immersion 25× objective and were analyzed with FIJI software. For each individual, images were acquired from two different slices, and within every slice, two different fields were imaged.

### Dendritic spine labelling

Dendritic spines were labelled with a protocol adapted from previous descriptions (Kim et al, 2007). Coronal brain slices (300 $\mu m$ thick) were obtained as described in the electrophysiology section. After incubation with the recovery solution, slices were incubated in aCSF for 1 h at 32°C and continuous bubbling (5% $CO_2$, 95% $O_2$). After that, the brain slices were gently placed on a glass slide and regions of interest were marked with a glass micropipette previously immersed in 10 $\mu M$ DiI (Thermo Fisher Scientific)/DMSO. The brain slices were then further incubated in aCSF for 1 h at 32°C under continuous bubbling (5% $CO_2$, 95% $O_2$), fixed for 10 min in 4% PFA at room temperature, washed three times in PBS, and mounted for microscopy analysis. Images were acquired in an LSM800 inverted microscope with a 100× objective, spines were manually counted, classified, and measured using the software FIJI. Head-free spines were considered *filopodia*; headed but neck-free were *stubby*; necked and with head smaller than 0.75 $\mu m^2$ were *thin*; necked and with head larger than 0.75 $\mu m^2$ were *mushroom*. Density and morphometric variables were quantified in three different fields per slice from three different brain slices for each individual.

### Amyloid plaque detection

The fluorescent pigment Thioflavin-S (Th-S) (Merck) was used to detect the deposition of amyloid plaques. Th-S has a peak of excitation and maximum emission of 430 and 550 nm, respectively. Brain sections were incubated with 0.1% Th-S in 50% ethanol for 10 min. After two washes in 50% ethanol and one wash in distilled water, sections were mounted using Mowiol Mounting Medium. Images were acquired using an Axiovert200 inverted microscope (Zeiss) coupled to a sCMOS monochrome camera (Excitation 461–488 nm, Emission 499–530 nm) with a dry 2.5× objective and were analyzed with FIJI software.

### Quantification of a$\beta$ in the cerebral cortex

One fourth of the cerebral cortex was homogenized in homogenization buffer (HB; 10 mg of tissue/100 $\mu l$ HB) containing 50 mM

Tris–HCl, 150 mM NaCl, 2 mM EDTA, and protease inhibitor; the pH was adjusted to 7.6. After centrifuging the homogenates at 200,000*g* and 4°C for 20 min, 5 *μ*l of the supernatant was used to determine the protein concentration by BCA assay and the rest was analyzed by specific ELISA assays against the A*β* peptide (soluble fraction). The resulting pellet of the centrifugation was resuspended in 100 *μ*l of 6 M guanidine-HCl, prepared in HB buffer, with the help of a sonicator (30 s at 10% amplitude) and a vortex. After an incubation period of 60 min at 25°C, the samples were centrifuged at 200,000*g* and 4°C for 20 min. The supernatant was transferred to a new tube and diluted 12× with HB buffer to reduce the Gu-HCl concentration to 0.5 M (insoluble fraction). 5 *μ*l of the insoluble fraction was used for the determination of the protein concentration by BCA assay and the rest was analyzed by specific ELISA assays.

Quantification of soluble and aggregated A*β*42 and A*β*40 in the cerebral cortex of WT and hAPP NL-F mice was performed in the soluble and insoluble fractions with a *β*-amyloid (42) ELISA Kit (290-62601; FUJIFILM Wako Pure Chemical Corporation) and a *β*-amyloid (40) ELISA Kit II (294-64701; FUJIFILM Wako Pure Chemical Corporation), respectively, following the instructions of the manufacturer. When required, samples were diluted in homogenization buffer. A standard curve with the following A*β*42 or A*β*40 concentrations was built to determine the A*β*42 or the A*β*40 levels in the samples: 100, 50, 25, 10, 5, 2.5, 1, and 0 pM. Oxidized TMB substrate is detected by reading the absorbance at 450 nm in a FLUOstar OPTIMA microplate reader (BMG LABTECH).

### RNA sequencing

Library preparation and sequencing RNA extracts were obtained from cortical samples using the Maxwell 16 LEV simplyRNA Tissue Kit (AS1280; Promega). Isolated RNA was then quantified using a NanoDrop ND-1000 (Thermo Fisher Scientific) at 260 nm absorbance, and the RNA integrity number was assessed by using a RNA 6000 Nano Bioanalyzer 2100 (Agilent).

Libraries were prepared from 3 biological replicates of each condition using the Truseq Stranded mRNA Library Prep Kit and sequenced using the HiSeq 2500 sequencing platform at the Genomics Unit of The Centre for Genomic Regulation (Barcelona) (CRG).

### RNA-sequencing data analysis

Raw reads were aligned against *M. musculus* genome assembly GRCm38 (mm10) using HISAT2 aligner (v.2.1.0) (Kim et al, 2015) using the –rna-strandness RF option. Aligned BAM files were sorted by name and indexed using samtools (v.1.9). Gene-level counts were generated with HTSeq-Count (v.0.11.2) (Anders et al, 2015) using the GENCODE (https://www.gencodegenes.org/) GTF file for gene coordinates with the following options: –stranded=reverse –order-=name –mode=intersection-nonempty –type=gene –idattr=gene_id. All gene expression data from the RNA-seq experiment were normalized using the median of ratios method in the DESeq2 R package (Love et al, 2014). DEGs between each group were obtained using DESeq2. Genes with q-values <0.05 were selected as DEGs. Finally, heatmaps were drawn using the pheatmap R package (version 1.0.12) to perform hierarchical clustering of the DEGs.

### Pathway enrichment analysis

An ORA (Boyle et al, 2004) using the ClusterProfiler R package (Yu et al, 2012) was conducted to determine whether known biological functions or processes were overrepresented in the experimentally-derived gene list. To report Gene Ontology (GO) terms on enrichment tests, a hypergeometric distribution was used and *P*-values were corrected for multiple testing by the Benjamini–Hochberg algorithm. Gene enrichment analysis visualization (cnetplot) was performed using the enrichplot (https://yulab-smu.top/biomedical-knowledge-mining-book/) and ggplot2 (https://ggplot2.tidyverse.org) R packages. The cnetplot organizes enriched terms into a network with edges connecting overlapping gene sets.

### Statistical analysis

The number of individuals used per experimental condition and experimental repeats is indicated in the figure legends. For electrophysiological experiments, n represents number of acute slices; N represents the number of individual mice. Data are presented as mean ± SEM. In all bar plots, individual data points are presented except for Fig 7C and E for visualization clarity. Data were analyzed with GraphPad Prism 6 or Excel 2011 software. For comparisons of the four experimental groups, one-way ANOVA or Kruskal–Wallis (when data were not normally distributed) was used. Tukey's multiple comparisons test or Dunn's multiple comparisons test were used for post hoc analysis of data tested with one-way ANOVA or Kruskal–Wallis, respectively. To focus on differences induced by the treatment, intra genotype analysis was performed using two-tailed unpaired *t* tests or two-tailed Mann–Whitney tests (when data were not normally distributed). For some of the figures, both analyses are reported: ANOVA and unpaired *t* test (significance indicated by * and # respectively). For Fig 7C and E, paired *t* tests were used. *P*-values < 0.05 were considered significant. Whenever possible, exact *P*-values are reported in the figure legends.

## Data Availability

All data supporting the findings of this study are available from the corresponding authors upon request. The Illumina paired end reads (FASTQ) generated for this study are available at The European Nucleotide Archive (ENA; http://www.ebi.ac.uk/ena/). Cortex raw data have been deposited under study accession number PRJEB61249.

## Supplementary Information

## Acknowledgements

The next-generation sequencing data analysis has been performed by the Genomics and next-generation sequencing Core Facility at the Centro de

Biología Molecular Severo Ochoa (CBM, CSIC-UAM), which is part of the CEI UAM+CSIC, Madrid, Spain—http://www.cbm.uam.es/genomica. The confocal microscopy analysis was performed at the Confocal Microscopy Facility of CBM. We would also like to thank Esperanza López-Merino from Jose A Esteban's laboratory (CBM), for her advice with the electrophysiology experiments. Finally, we thank Mercedes Hernández del Cerro for assisting us with the genotyping of the hAPP NL/F colony. This work was partially supported by grant PID2019-104389RB-I00 funded by MCIN/AEI/10.13039/501100011033 and "ERDF A way of making Europe" and by the European Union NextGenerationEU/PRTR CSIC's Interdisciplinary Thematic Platform PTI+ NEURO-AGING, both to CG Dotti, Marie Skłodowska-Curie Actions—Individual Fellowship (T2DM and AD, EU 708152) to F Guix and EU JPND "EpiAD" Grant to A Frank García and CG Dotti. M Carús-Cadavieco is the recipient of a Juan de la Cierva Formación Postdoctoral Fellowship (FJC2018-036152-I, MCIN/AEI).

## Author Contributions

M Carús-Cadavieco: data curation, formal analysis, validation, investigation, visualization, methodology, and writing—original draft, review, and editing.

I Berenguer López: data curation, formal analysis, validation, investigation, visualization, methodology, and writing—original draft, review, and editing.

A Montoro Canelo: data curation, formal analysis, investigation, and visualization.

MA Serrano-Lope: data curation, formal analysis, investigation, and visualization.

S González-de la Fuente: data curation, software, formal analysis, validation, visualization, and methodology.

B Aguado: software, formal analysis, validation, and methodology.

A Fernández-Rodrigo: formal analysis and methodology.

TC Saido: resources.

A Frank García: funding acquisition and validation.

C Venero: validation, methodology, and intellectual input.

JA Esteban: validation, methodology, and intellectual input.

F Guix: conceptualization, data curation, formal analysis, supervision, validation, investigation, methodology, and writing—original draft, review, and editing.

CG Dotti: conceptualization, supervision, funding acquisition, validation, investigation, methodology, project administration, and writing—original draft, review, and editing.

## Conflict of Interest Statement

The authors declare that they have no conflict of interest.

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
