## [Reviewer comments · Life Science Alliance]

Life Science Alliance

Cognitive decline in diabetic mice predisposed to Alzheimer's disease is greater than in wild type

Marta Carus-Cadavieco, Inés Berenguer López, Alba Montoro Canelo, Miguel Serrano-Lope, Sandra González de la Fuente, Begoña Aguado, Alba Fernandez-Rodrigo, Takaomi Saido, Ana Frank García, César Venero, Jose Esteban, Francesc Guix, and Carlos Dotti

DOI: <https://doi.org/10.26508/lsa.202201789>

Corresponding author(s): Carlos Dotti, Centro de Biología Molecular Severo Ochoa and Francesc Guix, Centro de Biología Molecular Severo Ochoa; Department of Bioengineering, Institut Químic de Sarrià (IQS) - Universitat Ramon Llull (URL)

Review Timeline:

Submission Date:	2022-10-28
Editorial Decision:	2022-12-09
Revision Received:	2023-02-28
Editorial Decision:	2023-03-21
Revision Received:	2023-03-27
Accepted:	2023-03-28

Transaction Report:

December 9, 2022

Re: Life Science Alliance manuscript #LSA-2022-01789-T

Prof. Carlos G. Dotti
Centro de Biología Molecular Severo Ochoa
Molecular Neuropathology
Nicolás Cabrera 1
Madrid 28049
Spain

Dear Dr. Dotti,

Thank you for submitting your manuscript entitled "Cognitive decline in diabetic mice with predisposition to Alzheimer's disease, not in wild type" to Life Science Alliance. The manuscript was assessed by an expert reviewer, whose comments are appended to this letter. We invite you to submit a revised manuscript addressing the Reviewer comments.

When submitting the revision, please include a letter addressing the reviewer's comments point by point.

Thank you for this interesting contribution to Life Science Alliance. We are looking forward to receiving your revised manuscript.

Sincerely,

B. MANUSCRIPT ORGANIZATION AND FORMATTING:

Reviewer #1 (Comments to the Authors (Required)):

This manuscript presents data on the effect of high fat diet [HFD] and elevated Abeta on cognitive function, neuropathology, circuit integrity, and gene expression changes. The humanized APP NL/F mouse model was utilized, compared to wildtype [WT]. The data presented supports the notion that many of these parameters are changed for the worse; however, methodology and data analysis details are lacking that prevent a rigorous evaluation of the data. Most significantly, statistical analysis methods and rationale are lacking as are methodological details regarding quantification of such things as immunoblot, neurobehavioral assays, immunohistochemistry, etc.

Specific comments:

Abstract: "tested the hypothesis that dementia in individuals with T2D is the consequence of carrying sub-clinical forms of dementia" is an hypothesis that cannot be tested as one cannot observe subclinical dementia. It is noted that inducing T2D in the hAPP NL/F mouse models accelerates the appearance of AD-like pathology; however, humans with these APP mutations will develop AD dementia. The present study observes what many previous studies have reported: comorbidities exacerbate AD pathology in rodent models of AD.

Materials and Methods: The manuscript could be shortened by removing experimental details from the Results section and bolstering the M&M section. Lack of methodological details in the M&M section forces the reader to fill in gaps from Results. Still, this does not fulfill all the missing experimental details. For example, immunoblot [the term should be used instead of western blot] descriptions do not specify if samples were pooled or if individual animals' samples were run. How was tissue weight:homogenization buffer volume controlled/normalized? Vague terms used. For example, it is stated that neurobehavior testing was performed 'the next day'. Was this exactly 24hrs or were animals tested at varying degrees of hours post training? Likewise, 'animals were acclimated for several days' - was it the same number of days for ALL test animals? Distance travelled in both OF and Barnes maze are important parameters to include in the data collection. Supplementary figure methods are missing.

Results: Again, many experimental details should be removed from this section and placed into M&M section to enhance clarity of the methodology. Several statements in the results section are not reflected in the figures. For example, 'T2DM impairs spatial learning and memory to a slightly larger extent in hAPP NL/F' - this is not indicated in figure 3. Results for Barnes maze indicate that distal cues were not utilized by the mouse subjects [panel 3E] in which a spatial [triangulation] search strategy was clearly not employed by all genotype/treatment groups. Likewise, no description of what or where these distal cues were in the training and testing room. Figures, results, as well as in M&M section, should be reported in order of their execution. Distance travelled should be reported, especially for OF, as this serves as a control for alternation in which it was noted that, overall, hAPP mice did not alternate to the same extent as WT. Barnes maze data reporting should include errors for attempts to enter a dummy escape hole. Statements regarding brain insulin resistance should be based on glucose uptake assays rather than phosphorylation status of the IR complex. Immunoblot loading controls are overexposed which calls into question normalization procedure. Other bands on immunoblots are also overexposed which precludes between group comparisons. Exposures should be in the linear range of the primary/secondary antibody combination.

Figures: What was the rationale for within genotype analysis rather than ANOVA and post hoc pairwise comparison of 4 groups? Thus, figures do not test whether hAPP performs differently from WT.

Discussion: Unable to assess as methodology and data analysis is incomplete.

Overall, this is a manuscript that provides moderately impactful advancement of the field. Critiques require major revision and resubmission to determine scientific integrity for ultimate publication.

Reviewer Query/comment:

This manuscript presents data on the effect of high fat diet [HFD] and elevated Abeta on cognitive function, neuropathology, circuit integrity, and gene expression changes. The humanized APP NL/F mouse model was utilized, compared to wildtype [WT]. The data presented supports the notion that many of these parameters are changed for the worse; however, methodology and data analysis details are lacking that prevent a rigorous evaluation of the data. Most significantly, statistical analysis methods and rationale are lacking as are methodological details regarding quantification of such things as immunoblot, neurobehavioral assays, immunohistochemistry, etc.

Authors' answer

We thank the reviewer for taking the time to evaluate our work. Following all the reviewer's suggestions, in this version the behavioral data has been re-evaluated with new statistical tests. In addition, we have reorganized the data, so that they are presented in a more logical way, with experimental details of the different tests in the materials and methods section, removing a substantial part from the results section. Third, we have removed sections of negative data (but we haven't removed the negative results, these are now embedded within other sections). We have also shortened and simplified the discussion. Each of these changes in response to reviewer questions are explained below, point by point.

Reviewer Query/comment:

Abstract:"tested the hypothesis that dementia in individuals with T2D is the consequence of carrying sub-clinical forms of dementia" is an hypothesis that cannot be tested as one cannot observe subclinical dementia. It is noted that

inducing T2D in the hAPP NL/F mouse models accelerates the appearance of AD-like pathology; however, humans with these APP mutations will develop AD dementia.

Authors' answer:

Subclinical is an illness that stays "below the surface" of clinical detection. A subclinical disease has no or minimally recognizable clinical findings. It is distinct from a clinical disease, which has signs and symptoms that can be more easily recognized. Regarding AD, all individuals with mutations in APP or PS will develop the clinical form yet they do have the subclinical form for many years, even decades. The mice utilized here have the subclinical form of AD since they present a series of mutations that lead to disease but do not present disease symptoms at the time of investigation. Therefore, our mice constitute, at the time we have studied them, a model of sub/clinical AD. In any case, to avoid miss-interpretations, we have removed the word sub-clinical from the abstract, replacing it with the expression predisposition.

The reviewer is totally right on stating that patients carrying APP mutations will develop AD dementia. However, it takes many years for patients with mutations in APP to manifest the first signs of dementia, even when mutations are present since birth. Similarly, it takes months (the equivalent to years in humans) for the hAPP NL/F mice to show signs of cognitive impairment. Thus, high A β levels are known to precede cognitive decline in humans for many years, and in a similar way, in hAPP NL/F mice, the high A β 42 levels precede the cognitive dysfunctions occurring in older animals. Since we carried out the analysis of hAPP NL/F mice at 12-14 months of age, before the appearance of cognitive deficits, our situation is equivalent to the preclinical situation in humans.

Reviewer Query/comment:

Materials and Methods: The manuscript could be shortened by removing experimental details from the Results section and bolstering the M&M section. Lack of methodological details in the M&M section forces the reader to fill in gaps from Results.

Authors' answer:

Following the reviewer's advice we have now moved most of experimental details previously in the results section to the materials and methods section, especially regarding the behavioral tests and the statistical analysis.

Reviewer Query/comment:

Still, this does not fulfill all the missing experimental details. For example, immunoblot [the term should be used instead of western blot] descriptions do not specify if samples were pooled or if individual animals' samples were run.

Authors' answer

We have now the term immunoblot instead of western blot in the manuscript, and extended the description of this assay in the materials and methods section. For all immunoblot experiments, individual animals' samples were run, so in the figures each point represents one individual from the corresponding experimental condition.

Reviewer Query/comment:

How was tissue weight:homogenization buffer volume controlled/normalized?
Vague terms used.

Authors' answer

Each cortical sample was homogenized with the same buffer volume. The protein concentration present in every homogenized sample was determined by using a Pierce™ BCA Protein Assay Kit (Thermo Scientific). Then, calculations were made to determine the necessary volume of each homogenized sample, and the necessary volume of distilled water to be mixed

together with Laemmli buffer. This way, all the immunoblot samples would have the same total protein amount (30 or 50 μg depending on the experiment) in the same final volume of 30 μl .

We now describe this in more detail in the materials and methods section (page 24).

Reviewer Query/comment:

For example, it is stated that neurobehavior testing was performed 'the next day'. Was this exactly 24hrs or were animals tested at varying degrees of hours post training?

Authors' answer:

We have clarified this in the manuscript. The time of analysis was 24 hs after the last test (page 27). We have also included in Supplementary Figure S2 a schematic representation of the experimental design for the Open Field and Novel Object (Supplementary Fig. S2A) and for the Barnes Maze (Supplementary Fig. S2D). We hope that these new panels will further improve the clarity of the methodology.

Reviewer Query/comment:

Likewise, 'animals were acclimated for several days' - was it the same number of days for ALL test animals?

Authors' answer:

This is now clarified in the revised text (page 26). Animals from all experimental conditions were simultaneously transferred to the behavioral room, so that all of them would have the same acclimation time.

Reviewer Query/comment:

Distance travelled in both OF and Barnes maze are important parameters to include in the data collection.

Authors' answer:

We have included the distance travelled in the OF and in the Barnes Maze (Fig. 3C and Supplementary Fig. S2E).

Reviewer Query/comment:

Supplementary figure methods are missing.

Authors' answer

In the corrected version, the materials and methods section describes all techniques and procedures employed both for the main and supplementary figures.

Reviewer Query/comment:

Results: Again, many experimental details should be removed from this section and placed into M&M section to enhance clarity of the methodology.

Authors' answer:

We have shortened the results section and extended the descriptions in the Materials and Methods.

Reviewer Query/comment:

Several statements in the results section are not reflected in the figures. For example, 'T2DM impairs spatial learning and memory to a slightly larger extent in hAPP NL/F' - this is not indicated in figure 3.

Authors' answer:

The revised text (page 9-11), figure 3 and the legend of the figure now reflect more clearly the differences.

Reviewer Query/comment:

Results for Barnes maze indicate that distal cues were not utilized by the mouse subjects [panel 3E] in which a spatial [triangulation] search strategy was clearly not employed by all genotype/treatment groups. Likewise, no

description of what or where these distal cues were in the training and testing room.

Authors' answer

We have extended the description of the Barnes Maze protocol in the materials and methods section (page 28-29). In short, extra-maze cues were positioned on the curtain backdrop surrounding the maze, at heights visible from the platform surface. These consisted of different colorful geometric shapes made of rigid paper, as described in (Gawel et al 2019; Venero, Ferraz 2017).

Figures, results, as well as in M&M section, should be reported in order of their execution.

Authors' answer

We apologize for the mistaken order. We have updated the report of the Open Field and the Novel Object, so that they appear in the order of their execution.

Reviewer Query/comment:

Distance travelled should be reported, especially for OF, as this serves as a control for alternation in which it was noted that, overall, hAPP mice did not alternate to the same extent as WT.

Authors' answer

We have now included the distance travelled during the Open Field test (Fig. 3C). T2DM decreased the distance travelled in NL/F mice, but not in Wild Type. Notably, the distance travelled in the OF by WT and NL/F mice under control diet was very similar. This reinforces the results obtained in the Y-Maze Spontaneous Alternations test, indicating that the decrease in alternations of hAPP mice is due to defects in short memory and not due to a general decrease in mobility.

Furthermore, the percentage of alternations for each mouse is calculated as the number of alternations made by the mouse divided by the total possible

alternations (total arm entries made by the animal minus 2) and multiplied by 100. In this way, this value is normalized and compensates for possible differences in the mice's movement through the Y-Maze.

It should be also noted, that a decrease in spontaneous alternations in hAPP compared to WT was already reported in (Saito 2014). Therefore it could be concluded that the genetic background of predisposition to AD impairs performance in the Y-Maze Spontaneous Alternations paradigm, and this is more exacerbated by T2DM.

Reviewer Query/comment:

Barnes maze data reporting should include errors for attempts to enter a dummy escape hole.

Authors' answer

We have now included primary errors made during the test day in the Barnes Maze (Supplementary Fig. S2F). This data show that T2DM increases the number of errors made by the mice before reaching the correct hole, in agreement with the worsening observed in the use of search strategies.

Reviewer Query/comment:

Statements regarding brain insulin resistance should be based on glucose uptake assays rather than phosphorylation status of the IR complex.

Authors' answer

In the revised manuscript we have removed the section regarding brain insulin resistance, and now report phosphorylation of IRS1 as Supplementary Figure S1, complementing the results of glucose uptake assays (page 8).

Reviewer Query/comment:

Immunoblot loading controls are overexposed which calls into question normalization procedure. Other bands on immunoblots are also overexposed

which precludes between group comparisons. Exposures should be in the linear range of the primary/secondary antibody combination.

Authors' answer

We took several photographs of the immunoblots at intervals of time to assure that the signal of bands was not saturated. Some of the bands show high intensity due to the high levels of these proteins in our samples, but at the moment of capturing our images, the intensity of the band had not reached its maximal level, and it is still possible to detect differences of intensity between bands within the same blot.

Reviewer Query/comment:

Figures: What was the rationale for within genotype analysis rather than ANOVA and post hoc pairwise comparison of 4 groups? Thus, figures do not test whether hAPP performs differently from WT.

Authors' answer:

The rationale for intra genotype characterization was very simple: does T2DM affect in the same way “normal” and “AD” individuals?. Therefore, in order to focus on the differences induced by the treatment, we decided to proceed with unpaired t-test and a within genotype analysis. However, for the studies in which the 4 experimental groups are present, we have performed ANOVA followed by post hoc comparisons of the 4 groups as the reviewer suggested. Statistical information has been updated in the figure legends, and also in the materials and methods section. For some of the figures, we report both analyses: ANOVA and unpaired t-test (significance indicated by * and # respectively), because we find that these differences revealed by within genotype analysis are interesting and complement the information provided by ANOVA

For Figure 4 and Supplementary Figure S3 (electrophysiological data), we have maintained the within genotype analysis, since the WT vs hAPP

comparison was already done in recent work from the lab (de Vidania 2020). Likewise, for Figure 2A and 2B (body weight and blood glucose levels) we have also maintained within genotype analysis, since we considered the WT vs hAPP comparison was not meaningful in this case.

.

March 21, 2023

RE: Life Science Alliance Manuscript #LSA-2022-01789-TR

Prof. Carlos G. Dotti
Centro de Biología Molecular Severo Ochoa
Molecular Neuropathology
Nicolás Cabrera 1
Madrid 28049
Spain

Dear Dr. Dotti,

Thank you for submitting your revised manuscript entitled "Cognitive decline in diabetic mice predisposed to Alzheimer's disease is greater than in wild type". We would be happy to publish your paper in Life Science Alliance pending final revisions necessary to meet our formatting guidelines.

- please address the remaining Reviewer comments
- please add the Twitter handle of your host institute/organization as well as your own or/and one of the authors in our system
- please make sure that each author that is listed in the manuscript is also entered in our manuscript portal
- ENA accession PRJEB41799 should be made publicly accessible at this point

Figure Check:

- please remove the panel A from the figure legend and the figure file for Figure S4; since this is the only panel for the figure, we don't need it designated

A. FINAL FILES:

B. MANUSCRIPT ORGANIZATION AND FORMATTING:

Sincerely,

Reviewer #1 (Comments to the Authors (Required)):

This resubmission addresses several of the previous criticisms. However, several issues remain.

The abstract states and therefore the study concludes that "chronic T2DM is not sufficient in itself to lead to severe cognitive disorders, but it makes them emerge in a predisposed background". This is not reflected in the data. FIG 3 show that that WT with induced T2D are impaired compared to their WT control cohort. Thus, T2D imposes cognitive deficits regardless of genetic risk for AD. This should be explained or the project conclusion tempered.

The justification for using the term 'subclinical' that is included in the rebuttal letter should be included in the intro and discussion, in some form.

Rationale for the statistical analyses employed in the rebuttal letter should be elucidated in the manuscript so the reader can judge for themselves if this is correctly applied.

Immunoblot methods should indicate how many times an individual animal sample was run for data analysis. Typically, samples are probed three times by running three separate immunoblots. Exposure time justification is lacking: as stated in the initial review, Exposures should be in the linear range of the primary/secondary antibody combination. Just because you find differences using the exposure time of choice, it does not determine that you are in the linear range for exposure time to detect the protein of choice with the antibody combinations used.

Again, the rebuttal letter contains more detail regarding the preparation of immunoblot samples than is in the manuscript. It should be the reverse.

Page 8: "defects in plasma membrane permeability" needs to be clarified mechanistically. Cell membranes per se do not change permeability. Ion channels do.

Manuscript needs editing for English language, grammar and spelling.

Editor Query (EQ)

-please address the remaining Reviewer comments

Answer (A): see below

EQ

-please add the Twitter handle of your host institute/organization as well as your own or/ and one of the authors in our system

A: added

EQ

-please make sure that each author that is listed in the manuscript is also entered in our manuscript portal

A: done

EQ

-ENA accession PRJEB41799 should be made publicly accessible at this point

A: We have already instructed ENA to open the access to the data. PRJEB41799 should be publicly accessible within the next days.

EQ:

-please remove the panel A from the figure legend and the figure file for Figure S4; since this is the only panel for the figure, we don't need it designated.

A: done

EQ:

LSA now encourages authors to provide a 30-60 second video where the study is briefly explained. We will use these videos on social media to promote the published paper and the presenting author (for examples, see

<https://twitter.com/LSAjournal/timelines/1437405065917124608>). Corresponding or first-authors are welcome to submit the video. Please submit only one video per manuscript. The video can be emailed to contact@life-science-alliance.org

A: video will be sent.

Reviewer query (RQ):

The abstract states and therefore the study concludes that "chronic T2DM is not sufficient in itself to lead to severe cognitive disorders, but it makes them emerge in a predisposed background". This is not reflected in the data. FIG 3 show that that WT with induced T2D are impaired compared to their WT control cohort. Thus, T2D imposes cognitive deficits regardless of genetic risk for AD. This should be explained or the project conclusion tempered.

A:

We have rewritten the final conclusion of the abstract to make it clear that the type of genetic background determines the severity of cognitive disorders in individuals with T2DM, which more correctly expresses what our results show (Highlighted text in the Abstract).

RQ:

The justification for using the term 'subclinical' that is included in the rebuttal letter should be included in the intro and discussion, in some form.

A:

We have included a paragraph in Introduction where it is explained why the mouse model used mimics the subclinical form of the disease (highlighted text, page 6).

RQ:

Rationale for the statistical analyses employed in the rebuttal letter should be elucidated in the manuscript so the reader can judge for themselves if this is correctly applied.

A:

This is now better explained in the Materials and Methods section (highlighted text, page 36-37)

RQ:

Immunoblot methods should indicate how many times an individual animal sample was run for data analysis. Typically, samples are probed three times by running three separate immunoblots.

A:

This is now explained in the Materials and Methods section (highlighted text, pages 24-25)

RQ:

Exposure time justification is lacking: as stated in the initial review.

A:

This is now explained in the Materials and Methods section (highlighted text, pages 24-25)

RQ:

Again, the rebuttal letter contains more detail regarding the preparation of immunoblot samples than is in the manuscript. It should be the reverse.

A:

A more detailed description of immunoblot preparation is now included (highlighted text, pages 24-25)

RQ:

Page 8: "defects in plasma membrane permeability" needs to be clarified mechanistically. Cell membranes per se do not change permeability. Ion channels do.

Answer:

We have now clarified that hAPP NL/F mice may present stronger phenotype than control mice due to differences in the levels of ion channels, possibly affecting trans-membrane permeability (highlighted text, Abstract, page 18 (Results) and page 20 (Discussion)).

RQ:

Manuscript needs editing for English language, grammar and spelling.

A:

We have had the manuscript corrected by a colleague whose native language is English.

March 28, 2023

RE: Life Science Alliance Manuscript #LSA-2022-01789-TRR

Prof. Carlos G. Dotti
Centro de Biología Molecular Severo Ochoa
Molecular Neuropathology
Nicolás Cabrera 1
Madrid 28049
Spain

Dear Dr. Dotti,

Thank you for submitting your Research Article entitled "Cognitive decline in diabetic mice predisposed to Alzheimer's disease is greater than in wild type". It is a pleasure to let you know that your manuscript is now accepted for publication in Life Science Alliance. Congratulations on this interesting work.

DISTRIBUTION OF MATERIALS:

Again, congratulations on a very nice paper. I hope you found the review process to be constructive and are pleased with how the manuscript was handled editorially. We look forward to future exciting submissions from your lab.

Sincerely,
